# Equal Long-term Benefit Rate: Adapting Static Fairness Notions to Sequential Decision Making

## Abstract

Decisions made by machine learning models may have lasting impacts over time, making long-term fairness a crucial consideration. It has been shown that when ignoring the long-term effect, naively imposing fairness criterion in static settings can actually exacerbate bias over time. To explicitly address biases in sequential decision-making, recent works formulate long-term fairness notions in Markov Decision Process (MDP) framework. They define the long-term bias to be the sum of static bias over each time step. However, we demonstrate that naively summing up the step-wise bias can cause a false sense of fairness since it fails to consider the importance difference of different time steps during transition. In this work, we introduce a long-term fairness notion called *Equal Long-term BEnefit RaTe* (`ELBERT`), which explicitly considers varying temporal importance and adapts static fairness principles to the sequential setting. Moreover, we show that the policy gradient of Long-term Benefit Rate can be analytically reduced to standard policy gradients. This makes standard policy optimization methods applicable for reducing bias, leading to our bias mitigation method `ELBERT`-PO. Extensive experiments on diverse sequential decision making environments consistently show that `ELBERT`-PO significantly reduces bias and maintains high utility.

## 1 Introduction

The growing use of machine learning in decision making systems has raised concerns about potential biases to different sub-populations from underrepresented ethnicity, race, or gender (Dwork et al., 2012). In the real-world scenario, the decisions made by these systems can not only cause immediate unfairness, but can also have long-term effects on the future status of different groups. For example, in a loan application decision-making case, excessively denying loans to individuals from a disadvantaged group can have a negative impact on their future financial status and thus exacerbate the unfair inferior financial status in the long run.

It has been shown that naively imposing static fairness constraints such as demographic parity (DP) (Dwork et al., 2012) or equal opportunity (Hardt et al., 2016) at every time step can actually exacerbate bias in the long run (Liu et al., 2018; D'Amour et al., 2020). To explicitly address biases in sequential decision making problems, recent works (Wen et al., 2021; Chi et al., 2021; Yin et al., 2023) formulate the long-term effects in the framework of Markov Decision Process (MDP). MDP models the dynamics through the transition of states, e.g. how the number of applicants and their financial status change at the next time step given the current decisions. Also, MDP allows leveraging techniques in reinforcement learning (RL) for finding policies with better utility and fairness.

In sequential decision-making, within a certain group, it is possible for some time steps to be more important than the others in terms of fairness. Considering such variation in temporal importance is crucial for long-term fairness. To illustrate this, consider a loan application scenario with two time steps and DP as the fairness criterion, as shown in Figure 1. For group blue, time step $t+1$ is more important than time step $t$, since the demand (number of loan applicants) is higher at $t+1$. For group red, time step $t$ is more important than $t+1$. For group blue, the bank provides a high $\frac{100}{100}$ acceptance rate on a more important time step $t+1$ and a low $\frac{0}{1}$ acceptance rate on a less important time step $t$. However, for group red, the bank supplies a low $\frac{0}{100}$ acceptance rate on a more important

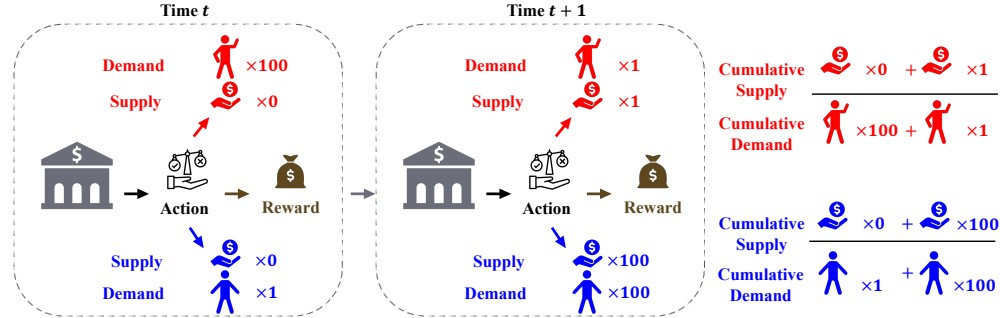

Figure 1: **(Left)** A loan application example where the bank decides whether to accept or reject the applicants from two groups in blue and red. At time step $t$, the bank approves 0 loans out of 1 qualified applicant from group blue and 0 loans out of 100 qualified applicants from group red. At time $t + 1$, the bank approves **100** loans out of **100** qualified applicants from group blue and **1** loan out of **1** qualified applicant from group red. **(Right)** The acceptance rate is 0 for both groups at time $t$ and is 1 for both groups at time $t + 1$. Therefore, the step-wise biases are zero and prior long-term fairness metrics lead to a false sense of fairness. In contrast, our proposed Long-term Benefit Rate calculates the bias as $|\frac{1}{101} - \frac{100}{101}|$ and successfully identifies the bias.

time step at $t$ and a high $\frac{1}{1}$ acceptance rate on a less important time step at $t + 1$. Therefore, group blue is more advantaged than group red, and bias emerges. In fact, the bank makes an overall $\frac{100}{101}$ acceptance rate for group blue, much higher than $\frac{1}{101}$ for group red.

However, variation in temporal importance is neglected in previous long-term fairness notions. For instance, Yin et al. (2023) define the long-term bias as the sum of step-wise bias (e.g. divergence of group acceptance rates), which is calculated as $(\frac{0}{1} - \frac{0}{100})^2 + (\frac{100}{100} - \frac{1}{1})^2 = 0$ in the aforementioned example. Another prior metric (Chi et al., 2021; Wen et al., 2021) defines the long-term bias as the difference of cumulative group fairness rewards (e.g. acceptance rates) between two groups, i.e. $(\frac{0}{1} + \frac{100}{100}) - (\frac{0}{100} + \frac{1}{1}) = 0$. In fact, both prior metrics claim no bias as long as step-wise biases are zero, which has been shown to potentially hurt long-term fairness both in prior work (Liu et al., 2018; D'Amour et al., 2020) and in the example above. Without considering the variation in temporal importance within a group, these prior metrics lead to a false sense of fairness.

In this work, we introduce Equal Long-term Benefit Rate (ELBERT), a long-term fairness criterion that adapts static fairness notions to sequential settings. Specifically, we define *Long-term Benefit Rate*, a general measure for the long-term well-being of a group, to be the ratio between the cumulative *group supply* (e.g. number of approved loans) and cumulative *group demand* (e.g. number of qualified applicants). For instance, in the loan application example, Long-term Benefit Rate calculates $\frac{100}{101}$ for group blue and $\frac{1}{101}$ for group red. By first summing up group supply and group demand separately and then taking the ratio, Long-term Benefit Rate takes into account that the group demand can change over time steps. Therefore, ELBERT explicitly accounts for the varying temporal importance during transition, eliminating the false sense of fairness induced by prior metrics. Moreover, ELBERT is a general framework that can adapt several static fairness notions to their long-term counterparts through customization of group supply and group demand.

Furthermore, we propose a principled bias mitigation method, ELBERT Policy Optimization (ELBERT-PO), to reduce the differences of Long-term Benefit Rate among groups. Note that optimizing Long-term Benefit Rate is challenging since it is not in the standard form of cumulative reward in RL and how to compute its policy gradient was previously unclear. To address this, we show that its policy gradient can be analytically reduced to the standard policy gradient in RL by deriving the *fairness-aware advantage function*, making commonly used policy optimization methods viable for bias mitigation. Experiments on diverse sequential decision making environments show that ELBERT-PO significantly improves long-term fairness while maintaining high utility.

**Summary of Contributions.** **(1)** We propose Equal Long-term Benefit Rate, which adapts static fairness criteria to sequential decision making. It explicitly considers change in temporal importance during transition, avoiding the false sense of fairness in prior notions. **(2)** We analytically show that standard policy optimization methods can be adapted for reducing bias, leading to our proposed ELBERT-PO. **(3)** Experiments on diverse sequential environments show that ELBERT-PO consistently achieves the lowest bias among all baselines while maintaining high reward.

## 2 ELBERT: EQUAL LONG-TERM BENEFIT RATE FOR LONG-TERM FAIRNESS

### 2.1 SUPPLY-DEMAND MARKOV DECISION PROCESS FOR LONG-TERM FAIRNESS

**Standard MDP.** A general sequential decision-making problem can be formulated as an MDP $\mathcal{M} = \langle \mathcal{S}, \mathcal{A}, \mu, T, R, \gamma \rangle$ (Sutton & Barto, 2018), where $\mathcal{S}$ is the state space (e.g. credit scores of applicants in the loan approval decision making mentioned above), $\mu$ is the initial state distribution, $\mathcal{A}$ is the action space (e.g. rejection or approval), $T : \mathcal{S} \times \mathcal{A} \to \Delta(\mathcal{S})$ is the transition dynamic, $R : \mathcal{S} \times \mathcal{A} \to \mathbb{R}$ is the immediate reward function (e.g. bank's earned profit) and $\gamma$ is the discounting factor. The goal of RL is to find a policy $\pi : \mathcal{S} \to \Delta(\mathcal{A})$ to maximize cumulative reward $\eta(\pi) := \mathbb{E}_\pi \left[ \sum_{t=0}^\infty \gamma^t R(s_t, a_t) \right]$, where $s_0 \sim \mu$, $a_t \sim \pi(\cdot|s_t)$, $s_{t+1} \sim T(\cdot|s_t, a_t)$ and $\gamma$ controls how myopic or farsighted the objective is.

Formulating fairness in MDP requires defining the long-term well-being of each group. This motivates us to rethink the static notions of group well-being and how to adapt them to MDP.

**Long-term group well-being: introducing supply and demand to MDPs.**

*(a) Supply and demand in static settings.* In many static fairness notions, the formulation of the group well-being can be unified as the ratio between supply and demand. For example, equal opportunity (EO) (Hardt et al., 2016) defines the well-being of group $g$ as $P[\hat{Y} = 1|G = g, Y = 1] = \frac{P[\hat{Y}=1,Y=1,G=g]}{P[Y=1,G=g]}$, where $\hat{Y} \in \{0,1\}$ is the binary decision (loan approval or rejection), $Y \in \{0,1\}$ is the target variable (repay or default) and $G$ is the group ID. The bias is defined as the disparity of the group well-being across different groups. In practice, given a dataset, the well-being of group $g$, using the notion of EO, is calculated as $\frac{S_g}{D_g}$, where the supply $S_g$ is the number of samples with $\{\hat{Y} = 1, Y = 1, G = g\}$ and the demand $D_g$ is the number of samples with $\{Y = 1, G = g\}$.

Note that such formulation in terms of supply and demand is not only restricted to EO, but is also compatible to other static fairness notions such as demographic parity (Dwork et al., 2012), equalized odds (Hardt et al., 2016), accuracy parity and equality of discovery probability (Elzayn et al., 2019), etc. We provide additional details in Appendix A.

*(b) Adapting to MDP.* In the sequential setting, each time step corresponds to a static dataset that comes with group supply and group demand. Therefore, to adapt them to MDP, we assume that in addition to immediate reward $R(s_t, a_t)$, the agent receives immediate group supply $S_g(s_t, a_t)$ and immediate group demand $D_g(s_t, a_t)$ at every time step $t$. This is formalized as the Supply-Demand MDP (SD-MDP) as shown in Figure 2 and defined as follows.

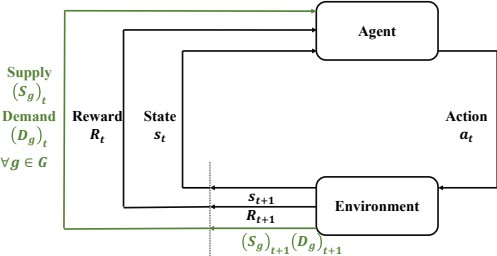

Figure 2: Supply Demand MDP (SD-MDP). In addition to the standard MDP (in black), SD-MDP returns group demand and group supply as fairness signals (in green).

**Definition 2.1** (Supply-Demand MDP (SD-MDP)). Given a group index set $G$ and a standard MDP $\mathcal{M} = \langle \mathcal{S}, \mathcal{A}, \mu, T, R, \gamma \rangle$, a Supply-Demand MDP is $\mathcal{M}_{\text{SD}} = \langle \mathcal{S}, \mathcal{A}, \mu, T, R, \gamma, \{S_g\}_{g \in G}, \{D_g\}_{g \in G} \rangle$. Here $\{S_g\}_{g \in G}$ and $\{D_g\}_{g \in G}$ are immediate group supply and group demand function for group $g$.

Compared with the standard MDP, in SD-MDP, an agent receives additional fairness signals $S_g(s_t, a_t)$ and $D_g(s_t, a_t)$ after taking action $a_t$ at each time step. To characterize the long-term group supply and group demand of a policy $\pi$, we define cumulative group supply and group demand as follows.

**Definition 2.2** (Cumulative Supply and Demand). Define the cumulative group supply as $\eta_g^S(\pi) := \mathbb{E}_\pi \left[ \sum_{t=0}^\infty \gamma^t S_g(s_t, a_t) \right]$ and cumulative group demand as $\eta_g^D(\pi) := \mathbb{E}_\pi \left[ \sum_{t=0}^\infty \gamma^t D_g(s_t, a_t) \right]$.

## 2.2 PROPOSED LONG-TERM FAIRNESS METRIC: EQUAL LONG-TERM BENEFIT RATE (ELBERT)

In the following definitions, we propose to measure the well-being of a group by the ratio of cumulative group supply and group demand and propose the corresponding fairness metric: Equal Long-term Benefit Rate (ELBERT).

**Definition 2.3** (Long-term Benefit Rate). Define the Long-term Benefit Rate of group $g$ as $\frac{\eta_g^S(\pi)}{\eta_g^D(\pi)}$. Define the bias of a policy as the maximal difference of Long-term Benefit Rate among groups, i.e., $b(\pi) = \max_{g \in G} \frac{\eta_g^S(\pi)}{\eta_g^D(\pi)} - \min_{g \in G} \frac{\eta_g^S(\pi)}{\eta_g^D(\pi)}$.

**RL with ELBERT.** Under the framework of ELBERT, the goal of reinforcement learning with fairness constraints is to find a policy to maximize the cumulative reward and keep the bias under a threshold $\delta$. In other words,

$$\max_{\pi} \eta(\pi) \quad \text{s.t.} \quad b(\pi) = \max_{g \in G} \frac{\eta_g^S(\pi)}{\eta_g^D(\pi)} - \min_{g \in G} \frac{\eta_g^S(\pi)}{\eta_g^D(\pi)} \leq \delta. \tag{1}$$

**Relationship with static fairness notions.** Note that in the special case when the length of time horizon is 1, Long-term Benefit Rate reduces to $\frac{S_g}{D_g}$, i.e., the static fairness notion.

**Versatility.** By choosing the proper definition of group supply $S_g$ and group demand $D_g$, ELBERT is customized to adapt the static notion to sequential decision-making. Section 5.1 shows how it covers commonly used fairness metrics in several real-world sequential decision-making settings.

**Comparison with other fairness notions in MDP.** One notion is return parity (Wen et al., 2021; Chi et al., 2022), which uses cumulative individual rewards to measure the group well-being for fairness consideration. It can be viewed as a special case of Long-term Benefit Rate with the demand function $D_g(s, a)$ being a constant function, which ignores the importance difference of different time steps during transition. Another potential notion is naively requiring zero bias at each time step. As demonstrated in Section 1, both notions can cause a false sense of fairness.

**Comparison with constrained RL.** The constraints in constrained RL are either step-wise (Wachi & Sui, 2020) or in the form of cumulative sum across time steps (Altman, 1999). Our constraint in Equation (1) considers all time steps but is not in the form of cumulative sum. Therefore, techniques in constrained RL cannot be direct applied. More detailed comparison is left to Appendix C.

## 3 ACHIEVING EQUAL LONG-TERM BENEFIT RATE

In this section, we will develop a bias mitigation algorithm, ELBERT Policy Optimization (ELBERT-PO) to solve the RL problem with the fairness considerations in Equation (1). In Section 3.1, we will formulate the training objective as a policy optimization problem and lay out the challenge of computing the policy gradient of this objective. In Section 3.2, we demonstrate how to compute the policy gradient of this objective by reducing it to standard policy gradient. In Section 3.3, we extend the objective and its solution to multi-group setting and deal with the non-smoothness of the maximum and minimum operator in Equation (1).

### 3.1 TRAINING OBJECTIVE AND ITS CHALLENGE

**Objective.** We first consider the special case of two groups $G = \{1, 2\}$, where Long-term Benefit Rate reduces to $|\frac{\eta_1^S(\pi)}{\eta_1^D(\pi)} - \frac{\eta_2^S(\pi)}{\eta_2^D(\pi)}|$. To solve the constrained problem in Equation (1), we propose to solve the unconstrained relaxation of it by maximizing the following objective:

$$J(\pi) = \eta(\pi) - \alpha b(\pi)^2 = \eta(\pi) - \alpha \left( \frac{\eta_1^S(\pi)}{\eta_1^D(\pi)} - \frac{\eta_2^S(\pi)}{\eta_2^D(\pi)} \right)^2 \tag{2}$$

where the bias coefficient $\alpha$ is a constant controlling the trade-off between the return and the bias.

**Challenge: policy gradient of $b(\pi)$.** To optimize the objective above, it is natural to use policy optimization methods that estimate the policy gradient and use stochastic gradient ascent to directly

improve policy performance. However, in order to compute the policy gradient $\nabla_\pi J(\pi)$ of $J(\pi)$ in Equation (2), one needs to compute $\nabla_\pi \eta(\pi)$ and $\nabla_\pi b(\pi)$. Although the term $\nabla_\pi \eta(\pi)$ is a standard policy gradient that has been extensively studied in RL (Schulman et al., 2016), it was previously unclear how to deal with $\nabla_\pi b(\pi) = \nabla_\pi(\frac{\eta_1^S(\pi)}{\eta_1^D(\pi)} - \frac{\eta_2^S(\pi)}{\eta_2^D(\pi)})$. In particular, since $b(\pi)$ is not of the form of expected total return, one cannot directly apply Bellman Equation (Sutton & Barto, 2018) to compute $b(\pi)$. Therefore, it is unclear how to leverage standard policy optimization methods (Schulman et al., 2017; 2015) to the objective function $J(\pi)$.

## 3.2 Solution to the objective

In this section, we show how to apply existing policy optimization methods in reinforcement learning to solve the objective in Equation (2). This is done by analytically reducing the objective's gradient $\nabla_\pi J(\pi)$ to standard policy gradients.

**Reduction to standard policy gradients.** For the simplicity of notation, we denote the term $b(\pi)^2$ in Equation (2) as a function of Long-term Benefit Rate $\{\frac{\eta_g^S(\pi)}{\eta_g^D(\pi)}\}_{g\in G}$ as $b(\pi)^2 = h(\frac{\eta_1^S(\pi)}{\eta_1^D(\pi)}, \frac{\eta_2^S(\pi)}{\eta_2^D(\pi)})$, where $h(z_1, z_2) = (z_1 - z_2)^2$. Therefore, $J(\pi) = \eta(\pi) - \alpha h(\frac{\eta_1^S(\pi)}{\eta_1^D(\pi)}, \frac{\eta_2^S(\pi)}{\eta_2^D(\pi)})$. The following proposition reduces the objective's gradient $\nabla_\pi J(\pi)$ to standard policy gradients.

**Proposition 3.1.** *The gradient of the objective function $J(\pi)$ can be calculated as*

$$\nabla_\pi J(\pi) = \nabla_\pi \eta(\pi) - \alpha \sum_{g\in G} \frac{\partial h}{\partial z_g}(\frac{1}{\eta_g^D(\pi)}\nabla_\pi \eta_g^S(\pi) - \frac{\eta_g^S(\pi)}{\eta_g^D(\pi)^2}\nabla_\pi \eta_g^D(\pi)), \qquad (3)$$

*where $\frac{\partial h}{\partial z_g}$ is the partial derivative of $h$ w.r.t. its $g$-th coordinate, evaluated at $(\frac{\eta_1^S(\pi)}{\eta_1^D(\pi)}, \frac{\eta_2^S(\pi)}{\eta_2^D(\pi)})$.*

Therefore, in order to estimate $\nabla_\pi J(\pi)$, one only needs to estimate the expected total supply and demand $\eta_g^S(\pi), \eta_g^D(\pi)$ as well as the standard policy gradients $\nabla_\pi \eta_g^S(\pi), \nabla_\pi \eta_g^D(\pi)$.

**Advantage function for policy gradients.** It is common to compute a policy gradient $\nabla_\pi \eta(\pi)$ using $\mathbb{E}_\pi\{\nabla_\pi \log \pi(a_t|s_t)A_t\}$, where $A_t$ is the advantage function of the reward $R$ (Sutton & Barto, 2018). Denote the advantage functions of $R, \{S_g\}_{g\in G}, \{D_g\}_{g\in G}$ as $A_t, \{A_{g,t}^S\}_{g\in G}, \{A_{g,t}^D\}_{g\in G}$. We can compute the gradient of the objective function $J(\pi)$ using advantage functions as follows.

**Proposition 3.2.** *In terms of advantage functions, the gradient $\nabla_\pi J(\pi)$ can be calculated as $\nabla_\pi J(\pi) = \mathbb{E}_\pi\{\nabla_\pi \log \pi(a_t|s_t)A_t^{fair}\}$, where the fairness-aware advantage function $A_t^{fair}$ is*

$$A_t^{fair} = A_t - \alpha \sum_{g\in G} \frac{\partial h}{\partial z_g}(\frac{1}{\eta_g^D(\pi)}A_{g,t}^S - \frac{\eta_g^S(\pi)}{\eta_g^D(\pi)^2}A_{g,t}^D) \qquad (4)$$

The detailed derivation is left to Appendix B.1. In practice, we use PPO (Schulman et al., 2017) with the fairness-aware advantage function $A_t^{fair}$ to update the policy network. The resulting algorithm, `ELBERT` Policy Optimization (`ELBERT`-PO), is given in Algorithm 1. In particular, in line 11-13, the PPO objective $J^{CLIP}(\theta)$ is used, where $\hat{\mathbb{E}}_{\pi_\theta}$ denotes the empirical average over samples collected by the policy $\pi_\theta$ and $\epsilon$ is a hyperparameter for clipping.

## 3.3 Extension to multi-group setting

**Challenge: Non-smoothness in multi-group bias.** When there are multiple groups, the objective is $J(\pi) = \eta(\pi) - \alpha b(\pi)^2 = \eta(\pi) - \alpha(\max_{g\in G} \frac{\eta_g^S(\pi)}{\eta_g^D(\pi)} - \min_{g\in G} \frac{\eta_g^S(\pi)}{\eta_g^D(\pi)})^2$. However, the max and min operator can cause non-smoothness in the objective during training. This is because only the groups with the maximal and minimal Long-term Benefit Rate will affect the bias term and thus the gradient of it. This is problematic especially when there are several other groups with Long-term Benefit Rate close to the maximal or minimal values. The training algorithm should consider all groups and decrease all the high Long-term Benefit Rate and increase low ones.

---

**Algorithm 1** ELBERT Policy Optimization (ELBERT-PO)

---

1: **Input:** Group set $G$, bias trade-off factor $\alpha$, bias function $h$, temperature $\beta$ (if multi-group)
2: Initialize policy network $\pi_\theta(a|s)$, value networks $V_\phi(s), V_{\phi_g^S}(s), V_{\phi_g^D}(s)$ for all $g \in G$
3: **for** $k \leftarrow 0, 1, ...$ **do**
4:     Collect a set of trajectories $\mathcal{D} \leftarrow \{\tau_k\}$ by running $\pi_\theta$ in the environment, each trajectory $\tau_k$
    contains $\tau_k :\leftarrow \{(s_t, a_t, r_t, s_{t+1})\}, t \in [|\tau_k|]$
5:     Compute the cumulative rewards, supply and demand $\eta, \eta_g^S, \eta_g^D$ of $\pi_\theta$ using Monte Carlo
6:     **for** each gradient step **do**
7:         Sample a mini-batch from $\mathcal{D}$
8:         Compute advantages $A_t, A_{g,t}^S, A_{g,t}^D$ using the current value networks $V_\phi(s)$, $V_{\phi_g^S}(s)$,
        $V_{\phi_g^D}(s)$ and mini-batch for all $g \in G$
9:         Compute $\frac{\partial h}{\partial z_g}$ at $(\frac{\eta_1^S}{\eta_1^D}, \cdots, \frac{\eta_M^S}{\eta_M^D})$
10:        Compute the fairness-aware advantage function:

$$A_t^{\text{fair}} = A_t - \alpha \sum_{g \in G} \frac{\partial h}{\partial z_g}(\frac{1}{\eta_g^D} A_{g,t}^S - \frac{\eta_g^S}{(\eta_g^D)^2} A_{g,t}^D)$$

11:        $R_t(\theta) \leftarrow \pi_\theta(s_t, a_t)/\pi_{\theta_{\text{old}}}(s_t, a_t)$
12:        $J^{\text{CLIP}}(\theta) \leftarrow \hat{\mathbb{E}}_{\pi_\theta}[\min(R_t(\theta)A_t^{\text{fair}}, \text{clip}(R_t(\theta), 1-\epsilon, 1+\epsilon)A_t^{\text{fair}})]$
13:        Update the policy network $\theta \leftarrow \theta + \tau \nabla_\theta J^{\text{CLIP}}(\theta)$
14:        Fit $V_\phi(s), V_{\phi_g^S}(s), V_{\phi_g^D}(s)$ by regression on the mean-squared error

---

**Soft bias in multi-group setting.** To solve this, we replace the max and min operator in $b(\pi)$ with their smoothed version controlled by the temperature $\beta > 0$ and define the soft bias $b^{\text{soft}}(\pi)$:

$$b^{\text{soft}}(\pi) = \frac{1}{\beta} \log \sum_{g \in G} \exp(\beta \frac{\eta_g^S(\pi)}{\eta_g^D(\pi)}) - \frac{1}{-\beta} \log \sum_{g \in G} \exp(-\beta \frac{\eta_g^S(\pi)}{\eta_g^D(\pi)}) \tag{5}$$

The relationship between the exact and soft bias is characterized by the following proposition:

**Proposition 3.3** (Approximation property of the soft bias). *Given a policy $\pi$, the number of groups $M$ and the temperature $\beta$, $b(\pi) \leq b^{soft}(\pi) \leq b(\pi) + \frac{2\log M}{\beta}$.*

In other words, the soft bias is an upper bound of the exact bias and moreover, the quality of such approximation is controllable: the gap between the two decreases as $\beta$ increases and vanishes when $\beta \to \infty$. We provide the proof in Appendix B.2. Therefore, we maximize the following objective

$$J(\pi) = \eta(\pi) - \alpha b^{\text{soft}}(\pi)^2 = \eta(\pi) - \alpha \left[ \frac{1}{\beta} \log \sum_g \exp(\beta \frac{\eta_g^S(\pi)}{\eta_g^D(\pi)}) - \frac{1}{-\beta} \log \sum_g \exp(-\beta \frac{\eta_g^S(\pi)}{\eta_g^D(\pi)}) \right]^2 \tag{6}$$

Note that we can write $b^{\text{soft}}(\pi)^2 = h(\frac{\eta_1^S(\pi)}{\eta_1^D(\pi)}, \frac{\eta_2^S(\pi)}{\eta_2^D(\pi)}, ..., \frac{\eta_M^S(\pi)}{\eta_M^D(\pi)})$ where $h(z) = [\frac{1}{\beta} \log \sum_g \exp(\beta z_g) - \frac{1}{-\beta} \log \sum_g \exp(-\beta z_g)]^2$, $z = (z_1, \cdots, z_M)$ and $J(\pi) = \eta(\pi) - \alpha h(\frac{\eta_1^S(\pi)}{\eta_1^D(\pi)}, \frac{\eta_2^S(\pi)}{\eta_2^D(\pi)}, ..., \frac{\eta_M^S(\pi)}{\eta_M^D(\pi)})$. The formula of $\nabla_\pi J(\pi)$ in Proposition 3.2 still holds and the training pipeline still follows Algorithm 1.

## 4 RELATED WORK

**Fairness criterion in MDP.** A line of work has formulated fairness in the framework of MDP. D'Amour et al. (2020) propose to study long-term fairness in MDP using simulation environments and shows that static fairness notions can contradict with long-term fairness. Return parity (Chi et al., 2022; Wen et al., 2021) assumes that the long-term group benefit can be represented by the sum of group benefit at each time step. However, as illustrated in Section 1, this assumption is problematic since it ignores the importance difference among different time steps during transition.

Yin et al. (2023) formulate the long-term bias as the sum of static bias at each time steps, suffering from the same problem. Our proposed `ELBERT` explicitly considers the varying temporal importance through the SD-MDP. Another work (Yu et al., 2022) assumes that there exists a long-term fairness measure for each state and proposes A-PPO to regularize the advantage function according to the fairness of the current and the next state. However, the assumption of Yu et al. (2022) does not hold in general since for a trajectory, the long-term fairness depends on the whole history of state-action pairs instead of only a single state. Moreover, A-PPO regularizes the advantage function to encourage the bias of the next time step to be smaller than the current one, without considering the whole future. However, our proposed `ELBERT`-PO considers the bias in all future steps, achieving long-term fairness in a principled way.

**Long-term fairness in other temporal models.** Long-term fairness is also studied in other temporal models. Liu et al. (2018) and Zhang et al. (2020b) show that naively imposing static fairness constraints in a one-step feedback model can actually harm the minority, showing the necessity of explicitly accounting for sequential decisions. Effort-based fairness (Heidari et al., 2019; Guldogan et al., 2022) measures bias as the disparity in the effort made by individuals from each group to get a target outcome, where the effort only considers one future time step. Long-term fairness has also been studied in multi-armed bandit (Joseph et al., 2016; Chen et al., 2020) and under distribution shifts in dynamic settings (Zhang et al., 2021; 2020a). In this work, we study long-term fairness in MDP since it is a general framework to model the dynamics in real world and allows leveraging existing RL techniques for finding high-utility policy with fairness constraints.

## 5 EXPERIMENT

In Section 5.1, we introduce the sequential decision making environments and their long-term fairness metrics. In particular, we explain how these metrics are covered by Long-term Benefit Rate via customizing group supply and demand. Section 5.2 demonstrates the effectiveness of `ELBERT`-PO on mitigating bias for two and multiple groups. In addition, the ablation study with varying values of the bias coefficient $\alpha$ is shown in Section 5.3.

### 5.1 ENVIRONMENTS AND THEIR LONG-TERM FAIRNESS CRITERIA

Following the experiments in Yu et al. (2022), we evaluate `ELBERT`-PO in three environments including **(1)** credit approval for lending (D'Amour et al., 2020) , **(2)** infectious disease control in population networks (Atwood et al., 2019) and **(3)** attention allocation for incident monitoring (D'Amour et al., 2020). To better examine the effectiveness of different methods, we modify the infectious disease and attention allocation environments to be more challenging. We give a brief introduction to each environment in the following. We leave the full description in Appendix D.1 and the experimental results on the original environment settings as in Yu et al. (2022) in Appendix D.3.

**Case 1: Lending.** In this environment, a bank decides whether to accept or reject loan applications and the applicants arrive one at a time sequentially. There are two groups among applicants ($G = \{1, 2\}$). The applicant at each time $t$ is from one of the groups $g_t$ and has a credit score sampled from the credit score distribution of group $g_t$. A higher credit score means higher repaying probability if the loan is approved. Group 2 is disadvantaged with a lower mean of the initial credit score distribution compared with Group 1. As for the dynamics, at time $t$, the credit score distribution of group $g_t$ shifts higher if its group member gets load approval (i.e. $\hat{Y}_t = 1$) and repays the loan (i.e. $Y_t = 1$). The immediate reward is the increment of the bank cash at each time step.

**Fairness criterion.** The bias is defined by $\left| \frac{\sum_t \mathbb{1}\{G_t=0, Y_t=\hat{Y}_t=1\}}{\sum_t \mathbb{1}\{G_t=0, Y_t=1\}} - \frac{\sum_t \mathbb{1}\{G_t=1, Y_t=\hat{Y}_t=1\}}{\sum_t \mathbb{1}\{G_t=1, Y_t=1\}} \right|$, which is the long-term extension of EO, where the group well-being is measured by the true positive rate.

**Case 2: Infectious disease control.** In this environment, the agent is tasked with vaccinating individuals within a social network to minimize the spread of a disease. The social network consists of individuals $V$ connected with the edges $E$, and each individual $v \in V$ is from one of the two groups $G = \{1, 2\}$. Every individual has a health state in being susceptible, infected or recovered, and the state space of the RL agent is given by the health state of all individuals. At each time step, the agent chooses no more than one individual to vaccinate. As for the dynamics, without vaccination, a susceptible individual gets infectious with probability that depends on the number of

infectious neighbors and a infected individual recovers with certain probability. When receiving the vaccine, the individual directly transit to recovered. Also, a recovered individual has certain probability to transit to being susceptible again. The immediate reward is the percentage of individuals in susceptible and recovered states in the whole network.

**Fairness criterion.** The fairness criterion is defined as $\left| \frac{\sum_t \text{vaccinations given}_{1t}}{\sum_t \text{newly infected}_{1t}} - \frac{\sum_t \text{vaccinations given}_{2t}}{\sum_t \text{newly infected}_{2t}} \right|$ where vaccinations given$_{gt}$ and newly infected$_{gt}$ are the number of vaccinations given to individuals from group $g$ and the number of new infected individuals from group $g$ at time $t$.

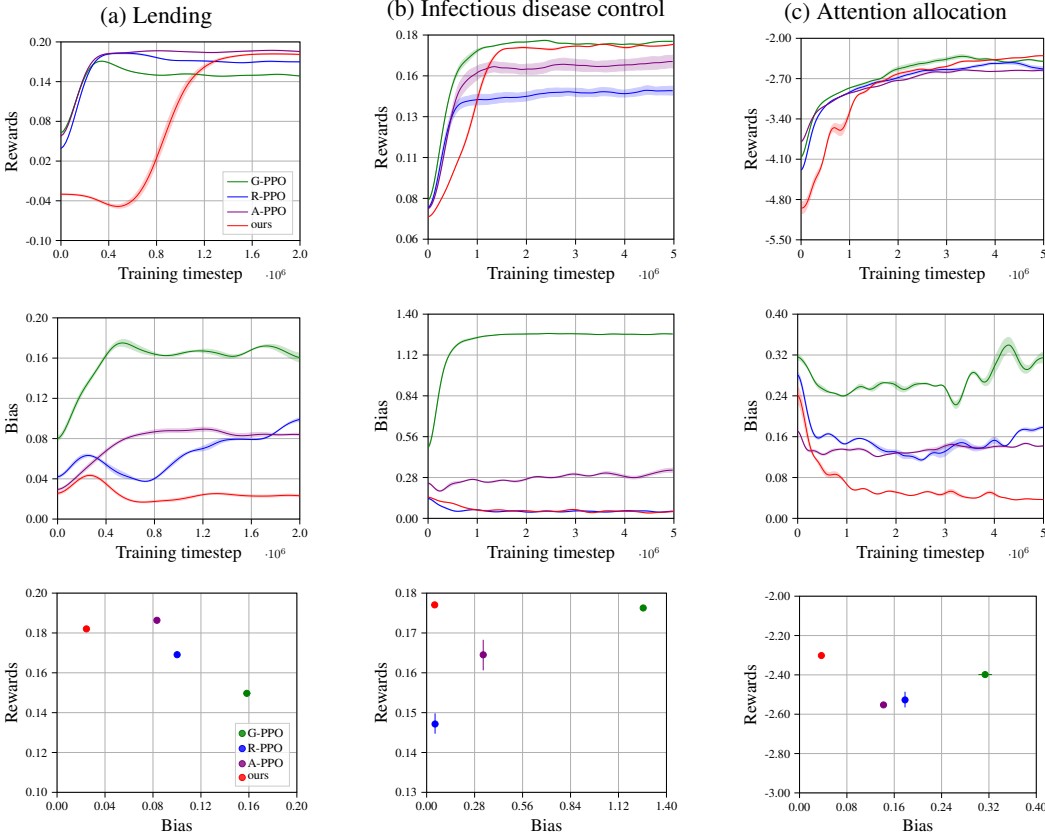

Figure 3: Reward and bias of `ELBERT`-PO (ours) and three other baselines (A-PPO, G-PPO, and R-PPO) in three environments (lending, infectious disease control and attention allocation). Each column shows the results in one environment. The third row shows the average reward versus the average bias, where `ELBERT`-PO consistently appears at the upper-left corner.

**Case 3: Attention allocation.** In this environment, the agent's task is to allocate 30 attention units to 5 sites (groups) to discover incidents, and each site has different initial incident rates. The agent's action is $a_t = \{a_{g,t}\}_{g=1}^5$, where $a_{g,t}$ is the number of allocated attention units for group $g$. The number of incidents $y_{g,t}$ is sampled from $\text{Poisson}(\mu_{g,t})$ with incident rate $\mu_{g,t}$ and the number of discovered incident is $\hat{y}_{g,t} = \min(a_{g,t}, y_{g,t})$. As for the dynamics, the incident rate changes according to $\mu_{g,t+1} = \mu_{g,t} - \underline{d}_g \cdot a_{g,t}$ if $a_{g,t} > 0$ and $\mu_{g,t+1} = \mu_{g,t} + \overline{d}_g$ otherwise, where the constants $\underline{d}_g$ and $\overline{d}_g$ are the dynamic rates for reduction and growth of the incident rate of group $g$. The agent's reward is $R(s_t, a_t) = -\sum_g (y_{g,t} - \hat{y}_{g,t})$, i.e., the negative sum of the missed incidents.

**Fairness criterion.** The group well-being is defined as the ratio between the total number of discovered incidents over time and the total number of incidents, and thus the bias is defined as $\max_{g \in G} \frac{\sum_t \hat{y}_{g,t}}{\sum_t y_{g,t}} - \min_{g \in G} \frac{\sum_t \hat{y}_{g,t}}{\sum_t y_{g,t}}$.

**Metrics as special cases of `ELBERT`.** All fairness criteria above used by prior works (Yu et al., 2022; D'Amour et al., 2020; Atwood et al., 2019) are covered by the general framework of `ELBERT` as special cases via customizing group supply and demand. For example, in the lending case, group

supply $D_g(s_t, a_t) = \mathbb{1}\{G_t = g, Y_t = 1\}$ and group demand $S_g(s_t, a_t) = \mathbb{1}\{G_t = g, Y_t = \hat{Y}_t = 1\}$. Therefore, ELBERT-PO can be used as a principled bias mitigation method for all of these environments, which is demonstrated in the next section.

## 5.2 EFFECTIVENESS OF ELBERT-PO

**Baselines.** Following Yu et al. (2022), we consider the following RL baselines. (1) A-PPO (Yu et al., 2022), which regularizes the advantage function to decrease the bias of the next time steps but does not consider the biases in all future steps. (2) Greedy PPO (G-PPO), which greedily maximizes reward without any fairness considerations. (3) Reward-Only Fairness Constrained PPO (R-PPO), a heuristic method which injects the historical bias into the immediate reward. In particular, it adds $-\max(0, \Delta_t - \omega)$ to the immediate reward $R_t$ at time $t$, where $\Delta_t$ is the overall bias of all previous time steps and $\omega$ is a threshold. The hyperparameters of all methods are given in Appendix D.2.

**Results: ELBERT-PO consistently achieves the lowest bias while maintaining high reward.** The performance of ELBERT-PO and baselines are shown in Figure 3. *(1) Lending.* ELBERT-PO achieves the lowest bias of $0.02$, significantly decreasing the bias of G-PPO by $87.5\%$, R-PPO and A-PPO by over $75\%$, while obtaining high reward. *(2) Infectious.* ELBERT-PO achieves the lowest bias of $0.01$ among all methods. Although R-PPO also achieves the same low bias, it suffers from much lower reward, indicating that directly injecting bias into immediate reward can harm reward maximization. A-PPO obtains a relatively large bias, suggesting that only considering the bias of the next time step can be insufficient for mitigating bias that involves the whole future time steps. Furthermore, ELBERT-PO obtains the same reward as G-PPO, higher than other bias mitigation baselines. *(3) Allocation.* ELBERT-PO achieves the lowest bias and the highest reward of among all methods. This shows the effectiveness of ELBERT-PO in the multi-group setting.

## 5.3 EFFECT OF THE BIAS COEFFICIENT

Figure 4 shows the learning curve with different values of the bias coefficient $\alpha$ in the attention environment. We observe that larger $\alpha$ leads to lower bias, and such effect is diminishing as $\alpha$ becomes larger. In terms of reward, we find that increasing $\alpha$ leads to slower convergence. This is expected since the reward signal becomes weaker as $\alpha$ increases. However, we find that larger $\alpha$ leads to slightly higher rewards. This sug-

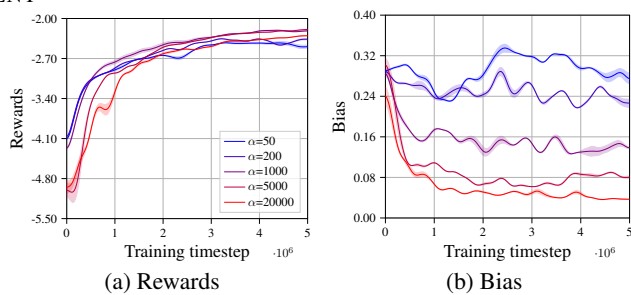

(a) Rewards      (b) Bias

Figure 4: Learning curve of ELBERT-PO on the attention allocation environment with different $\alpha$.

gests that lower bias does not necessarily leads to lower rewards, and learning with fairness consideration may help reward maximization. More results on $\alpha$ in other environments as well as how group supply and demand change during training for all methods can be found in Appendix D.3.

## 6 CONCLUSIONS AND DISCUSSIONS

In this work, we introduce Equal Long-term Benefit Rate (ELBERT) for adapting static fairness notions to sequential decision-making. It explicitly accounts for the varying temporal importance instead of naively summing up step-wise biases. For bias mitigation, we address the challenge of computing the policy gradient of Long-term Benefit Rate by analytically reducing it to the standard policy gradients through the fairness-aware advantage function, leading to our proposed ELBERT-PO. Experiments demonstrate that it significantly reduces bias while maintaining high utility.

One limitation is that ELBERT focuses on long-term adaptations of static fairness notions, which mainly consider the supply-demand ratio but not the demand itself. However, in real world applications, extra constraints on demand might be needed. For example, the demand should not be too large (e.g. when demand is the number of infected individuals) or too small (e.g. when demand is the number of qualified applicants). To address this, we show in Appendix E that ELBERT-PO also works when additional terms to regularize demand are incorporated in the objective function.

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

# A FAIRNESS NOTIONS WITH THE SUPPLY AND DEMAND FORMULATION

In this section, we demonstrate that in the static settings, the supply and demand formulation in Section 2 can cover many popular fairness notions. This means that the proposed Supply Demand MDP is expressive enough to extend several popular static fairness notions to the sequential settings. In the following, we give a list of examples to show, in the static setting, how to formulate several popular fairness criteria as the ratio between the supply and demand. For simplicity, we consider the agent's decision to be binary, though the analysis naturally extends to multi-class settings.

**Notations.** Denote $\hat{Y} \in \{0, 1\}$ as the binary decision (loan approval or rejection), $Y \in \{0, 1\}$ as the target variable (repay or default) and $G$ as the group ID.

**Demographic Parity.** The well-being of a group $g$ in Demographic Parity (DP) (Dwork et al., 2012) is defined as $P[\hat{Y} = 1 | G = g] = \frac{P[\hat{Y}=1, G=g]}{P[G=g]}$ and DP requires such group well-being to equalized among groups. In practice, given a dataset, the well-being of group $g$ is calculated as $\frac{S_g}{D_g}$, where the supply $S_g$ is the number of samples with $\{\hat{Y} = 1, G = g\}$ (e.g. the number of accepted individuals in group $g$) and the demand $D_g$ is the number of samples with $\{G = g\}$ (e.g. the total number of individuals from group $g$).

**Equal Opportunity.** The well-being of a group $g$ in Equal Opportunity (EO) (Dwork et al., 2012) is defined as $P[\hat{Y} = 1 | G = g, Y = 1] = \frac{P[\hat{Y}=1, Y=1, G=g]}{P[Y=1, G=g]}$ and EO requires such group well-being to equalized among groups. In practice, given a dataset, the well-being of group $g$ is calculated as $\frac{S_g}{D_g}$, where the supply $S_g$ is the number of samples with $\{\hat{Y} = 1, Y = 1, G = g\}$ (e.g. the number of qualified and accepted individuals in group $g$) and the demand $D_g$ is the number of samples with $\{Y = 1, G = g\}$ (e.g. the number of qualified individuals from group $g$).

**Equality of discovery probability: a special case of EO.** Equality of discovery probability (Elzayn et al., 2019) requires that the discovery probability to be equal among groups. For example, in predictive policing setting, it requires that conditional on committing a crime ($Y = 1$), the probability that an individual is apprehended ($\hat{Y} = 1$) should be independent of the district ID (group ID) $g$. This is a special case of EO in specific application settings.

**Equalized Odds.** Equalized Odds (Dwork et al., 2012) requires that both the True Positive Rate (TPR) $P[\hat{Y} = 1 | G = g, Y = 1] = \frac{P[\hat{Y}=1, Y=1, G=g]}{P[Y=1, G=g]}$ and the False Positive Rate (FPR) $P[\hat{Y} = 1 | G = g, Y = 0] = \frac{P[\hat{Y}=1, Y=0, G=g]}{P[Y=0, G=g]}$ equalize among groups. In practice, given a dataset, **(a)** the TPR of group $g$ is calculated as $\frac{S_g^T}{D_g^T}$, where the supply $S_g^T$ is the number of samples with $\{\hat{Y} = 1, Y = 1, G = g\}$ (e.g. the number of qualified and accepted individuals in group $g$) and the demand $D_g^T$ is the number of samples with $\{Y = 1, G = g\}$ (e.g. the number of qualified individuals from group $g$). **(b)** The FPR of group $g$ is calculated as $\frac{S_g^F}{D_g^F}$, where the supply $S_g^F$ is the number of samples with $\{\hat{Y} = 1, Y = 0, G = g\}$ (e.g. the number of unqualified but accepted individuals in group $g$) and the demand $D_g^F$ is the number of samples with $\{Y = 0, G = g\}$ (e.g. the number of unqualified individuals from group $g$).

**Extending Equalized Odds to sequential settings using SD-MDP.** The long-term adaption of Equalized Odds can be included by the Supply Demand MDP via allowing it to have two sets of supply-demand pairs: for every group $g$, $(D_g^T, S_g^T)$ and $(D_g^F, S_g^F)$. In particular, define the cumulative supply and demand for both supply-demand pairs: the cumulative group supply for TPR $\eta_g^{S,T}(\pi) := \mathbb{E}_\pi \left[ \sum_{t=0}^\infty \gamma^t S_g^T(s_t, a_t) \right]$ and cumulative group demand for TPR as $\eta_g^{D,T}(\pi) := \mathbb{E}_\pi \left[ \sum_{t=0}^\infty \gamma^t D_g^T(s_t, a_t) \right]$. The cumulative group supply for FPR $\eta_g^{S,F}(\pi) := \mathbb{E}_\pi \left[ \sum_{t=0}^\infty \gamma^t S_g^F(s_t, a_t) \right]$ and cumulative group demand for FPR as $\eta_g^{D,F}(\pi) := \mathbb{E}_\pi \left[ \sum_{t=0}^\infty \gamma^t D_g^F(s_t, a_t) \right]$. Since the bias considers both TPR and FPR, we define the bias for both:

$b^T(\pi) = \max_{g \in G} \frac{\eta_g^{S,T}(\pi)}{\eta_g^{D,T}(\pi)} - \min_{g \in G} \frac{\eta_g^{S,T}(\pi)}{\eta_g^{D,T}(\pi)}$ and $b^F(\pi) = \max_{g \in G} \frac{\eta_g^{S,F}(\pi)}{\eta_g^{D,F}(\pi)} - \min_{g \in G} \frac{\eta_g^{S,F}(\pi)}{\eta_g^{D,F}(\pi)}$.
The goal of RL with Equalized Odds constraints can be formulated as

$$\max_{\pi} \eta(\pi)$$

$$\text{s.t. } b^T(\pi) = \max_{g \in G} \frac{\eta_g^{S,T}(\pi)}{\eta_g^{D,T}(\pi)} - \min_{g \in G} \frac{\eta_g^{S,T}(\pi)}{\eta_g^{D,T}(\pi)} \leq \epsilon \tag{7}$$

$$b^F(\pi) = \max_{g \in G} \frac{\eta_g^{S,F}(\pi)}{\eta_g^{D,F}(\pi)} - \min_{g \in G} \frac{\eta_g^{S,F}(\pi)}{\eta_g^{D,F}(\pi)} \leq \epsilon.$$

In practice, we treat the hard constraints as regularization and use the following objective function

$$J(\pi) = \eta(\pi) - \alpha b^T(\pi)^2 - \alpha b^F(\pi)^2 \tag{8}$$

where $\alpha$ is a trade-off constant between return and fairness. The gradient $\nabla_\pi(J\pi)$ can still be computed using techniques presented in 3, since both bias terms $b^T(\pi)$ and $b^F(\pi)$ are still in the form of ratio between cumulative supply and demand.

**Accuracy Parity.** Accuracy Parity defines the well-being of group $g$ as $P[\hat{Y} = Y | G = g] = \frac{P[\hat{Y}=Y, G=g]}{P[G=g]}$, which is the accuracy of predicting $Y$ using $\hat{Y}$ among individuals from the group $g$. In practice, this is computed by $\frac{S_g}{D_g}$, where the supply $S_g$ is the number of samples with $\{\hat{Y} = Y, G = g\}$ (e.g. the number of individuals with correct predictions in group $g$) and the demand $D_g$ is the number of samples with $\{G = g\}$ (e.g. the total number of individuals from group $g$).

# B MATHEMATICAL DERIVATIONS

## B.1 FAIRNESS-AWARE ADVANTAGE FUNCTION

In this section, we show how to apply existing policy optimization methods to solve the objective in Equation (2). This is done by analytically reducing the policy gradient $\nabla_\pi b(\pi)$ of the bias to standard policy gradients.

**Gradient of the objective.** For the simplicity of notation, we denote the term $b(\pi)^2$ in Equation (2) as a function of Long-term Benefit Rate $\{\frac{\eta_g^S(\pi)}{\eta_g^D(\pi)}\}_{g \in G}$ as $b(\pi)^2 = h(\frac{\eta_1^S(\pi)}{\eta_1^D(\pi)}, \frac{\eta_2^S(\pi)}{\eta_2^D(\pi)})$, where $h(z_1, z_2) = (z_1 - z_2)^2$. Therefore, $J(\pi) = \eta(\pi) - \alpha h(\frac{\eta_1^S(\pi)}{\eta_1^D(\pi)}, \frac{\eta_2^S(\pi)}{\eta_2^D(\pi)})$. By chain rule, we can compute the gradient of the objective as follows.

$$\nabla_\pi J(\pi) = \nabla_\pi \eta(\pi) - \alpha \sum_{g \in G} \frac{\partial h}{\partial z_g} \nabla_\pi \left( \frac{\eta_g^S(\pi)}{\eta_g^D(\pi)} \right) \tag{9}$$

where $\frac{\partial h}{\partial z_g}$ is the partial derivative of $h$ w.r.t. its $g$-th coordinate, evaluated at $(\frac{\eta_1^S(\pi)}{\eta_1^D(\pi)}, \frac{\eta_2^S(\pi)}{\eta_2^D(\pi)})$. Note that $\nabla_\pi \eta(\pi)$ in Equation (9) is a standard policy gradient, whereas $\nabla_\pi(\frac{\eta_g^S(\pi)}{\eta_g^D(\pi)})$ is not.

**Reduction to standard policy gradient.** For $\nabla_\pi(\frac{\eta_g^S(\pi)}{\eta_g^D(\pi)})$, we apply the chain rule again as follows

$$\nabla_\pi \left( \frac{\eta_g^S(\pi)}{\eta_g^D(\pi)} \right) = \frac{1}{\eta_g^D(\pi)} \nabla_\pi \eta_g^S(\pi) - \frac{\eta_g^S(\pi)}{\eta_g^D(\pi)^2} \nabla_\pi \eta_g^D(\pi) \tag{10}$$

Therefore, in order to estimate $\nabla_\pi(\frac{\eta_g^S(\pi)}{\eta_g^D(\pi)})$, one only needs to estimate the expected total supply and demand $\eta_g^S(\pi), \eta_g^D(\pi)$ as well as the standard policy gradients $\nabla_\pi \eta_g^S(\pi), \nabla_\pi \eta_g^D(\pi)$.

**Advantage function for policy gradient.** It is common to compute a policy gradient $\nabla_\pi \eta(\pi)$ using $\mathbb{E}_\pi \{\nabla_\pi \log \pi(a_t|s_t) A_t\}$, where $A_t$ is the advantage function of the reward $R$ (Sutton & Barto, 2018). Denote the advantage functions of $R, \{S_g\}_{g \in G}, \{D_g\}_{g \in G}$ as $A_t, \{A_{g,t}^S\}_{g \in G}, \{A_{g,t}^D\}_{g \in G}$. $\nabla_\pi(\frac{\eta_g^S(\pi)}{\eta_g^D(\pi)})$ in Equation (10) can thus be written as

$$\nabla_\pi(\frac{\eta_g^S(\pi)}{\eta_g^D(\pi)}) = \mathbb{E}_\pi\left\{\nabla_\pi \log \pi(a_t|s_t)(\frac{1}{\eta_g^D(\pi)}A_{g,t}^S - \frac{\eta_g^S(\pi)}{\eta_g^D(\pi)^2}A_{g,t}^D)\right\} \qquad (11)$$

By plugging Equation (11) into Equation (9), we obtain the gradient of the objective $J(\pi)$ using advantage functions as follows

$$\nabla_\pi J(\pi) = \mathbb{E}_\pi\left\{\nabla_\pi \log \pi(a_t|s_t)\left[A_t - \alpha \sum_{g \in G}\frac{\partial h}{\partial z_g}(\frac{1}{\eta_g^D(\pi)}A_{g,t}^S - \frac{\eta_g^S(\pi)}{\eta_g^D(\pi)^2}A_{g,t}^D)\right]\right\} \qquad (12)$$

Therefore, $\nabla_\pi J(\pi) = \mathbb{E}_\pi\{\nabla_\pi \log \pi(a_t|s_t)A_t^{\text{fair}}\}$, where $A_t^{\text{fair}} = A_t - \alpha \sum_{g \in G}\frac{\partial h}{\partial z_g}(\frac{1}{\eta_g^D(\pi)}A_{g,t}^S - \frac{\eta_g^S(\pi)}{\eta_g^D(\pi)^2}A_{g,t}^D)$ is defined as the *fairness-aware advantage* function.

## B.2 RELATIONSHIP BETWEEN THE SOFT BIAS AND THE BIAS

We would like to show the mathematical relationship between the soft bias and bias, as shown in Theorem 3.3. This is done by analyzing the max and min operator as well as their soft counterparts through the log sum trick, which is also used in prior work (Xu et al., 2023). We restate the full proposition and present the proof below.

**Proposition B.1.** *Given a policy $\pi$, the number of groups $M$ and the temperature $\beta$, define the soft bias as*

$$b^{soft}(\pi) = \frac{1}{\beta}\log\sum_{g \in G}\exp(\beta\frac{\eta_g^S(\pi)}{\eta_g^D(\pi)}) - \frac{1}{-\beta}\log\sum_{g \in G}\exp(-\beta\frac{\eta_g^S(\pi)}{\eta_g^D(\pi)}).$$

*The bias is defined as*

$$b(\pi) = \max_{g \in G}\frac{\eta_g^S(\pi)}{\eta_g^D(\pi)} - \min_{g \in G}\frac{\eta_g^S(\pi)}{\eta_g^D(\pi)}.$$

*We have that*

$$b(\pi) \le b^{soft}(\pi) \le b(\pi) + \frac{2\log M}{\beta}.$$

*Proof.* First consider the first term $\frac{1}{\beta}\log\sum_{g \in G}\exp(\beta\frac{\eta_g^S(\pi)}{\eta_g^D(\pi)})$ in the soft bias $b^{\text{soft}}(\pi)$.

On the one hand, we have that

$$\frac{1}{\beta}\log\sum_{g \in G}\exp(\beta\frac{\eta_g^S(\pi)}{\eta_g^D(\pi)}) > \frac{1}{\beta}\log\exp(\beta\max_{g \in G}\frac{\eta_g^S(\pi)}{\eta_g^D(\pi)})$$
$$= \max_{g \in G}\frac{\eta_g^S(\pi)}{\eta_g^D(\pi)} \qquad (13)$$

On the other hand, we have that

$$\frac{1}{\beta}\log\sum_{g \in G}\exp(\beta\frac{\eta_g^S(\pi)}{\eta_g^D(\pi)}) \le \frac{1}{\beta}\log M\exp(\beta\max_{g \in G}\frac{\eta_g^S(\pi)}{\eta_g^D(\pi)})$$
$$= \max_{g \in G}\frac{\eta_g^S(\pi)}{\eta_g^D(\pi)} + \frac{\log M}{\beta} \qquad (14)$$

Therefore, $\max_{g \in G}\frac{\eta_g^S(\pi)}{\eta_g^D(\pi)} < \frac{1}{\beta}\log\sum_{g \in G}\exp(\beta\frac{\eta_g^S(\pi)}{\eta_g^D(\pi)}) \le \max_{g \in G}\frac{\eta_g^S(\pi)}{\eta_g^D(\pi)} + \frac{\log M}{\beta}$.

Similarly, it can be shown that $\min_{g \in G}\frac{\eta_g^S(\pi)}{\eta_g^D(\pi)} - \frac{\log M}{\beta} \le \frac{1}{-\beta}\log\sum_{g \in G}\exp(-\beta\frac{\eta_g^S(\pi)}{\eta_g^D(\pi)}) < \min_{g \in G}\frac{\eta_g^S(\pi)}{\eta_g^D(\pi)}$.

By subtracting the two, we conclude that $b(\pi) \le b^{\text{soft}}(\pi) \le b(\pi) + \frac{2\log M}{\beta}$.

$\square$

## C  CONNECTION TO CONSTRAINED RL

In this section, we compare our proposed `ELBERT` with the previous works of constrained Reinforcement Learning (RL). Prior formulations of constrained RL can be mainly categorized into two groups as follows. We will explain that neither of them can be directly applied to solve our fairness objective in Equation (1) in the `ELBERT` framework.

**Cumulative cost constraints**   The first category is learning a policy with cost constraints that are in the form of cumulative sum, usually known as constrained MDPs (CMDPs) (Altman, 1999). It is formulated as a tuple $\mathcal{M} = \langle \mathcal{S}, \mathcal{A}, \mu, T, R, C, \gamma \rangle$. In addition to the components in the standard MDP, there is an extra cost function $C : \mathcal{S} \times \mathcal{A} \to \mathbb{R}$. The feasible policy is subject to the cumulative cost under a threshold $\delta$. Mathematically, the goal is formulated as

$$\max_{\pi} \eta(\pi) \quad \text{s.t.} \quad \eta_C(\pi) = \mathbb{E}_{\pi} \left[ \sum_{t=0}^{\infty} \gamma^t C(s_t, a_t) \right] \leq \delta. \tag{15}$$

A series of works (Satija et al., 2020; Zhou et al., 2022) has studied the problem in Equation (15). Notably, methods for solving CMDPs rely on Bellman equation to evaluate the value function or the policy gradient of the cumulative cost. Specifically, the cost function in Equation (15) is similar to the reward in standard MDPs and thus the cumulative cost can be reformulated as the expectation of state value function of cost over states, i.e., $\eta_C(\pi) = \mathbb{E}_{s \sim \mu}[V_C^\pi(s)]$. Here the state value function

$$V_C^\pi(s) = \mathbb{E}_{\pi} \left[ \sum_{t=0}^{\infty} \gamma^t C(s_t, a_t) \middle| \pi, s_0 = s \right] \tag{16}$$

satisfies the Bellman equation

$$V_C^\pi(s) = \sum_{a} \pi(a|s) \sum_{s'} T(s'|s, a)[R(s, a) + \gamma V_C^\pi(s')] \tag{17}$$

which can be used to evaluate the value function or the policy gradients of the cumulative cost (Sutton & Barto, 2018).

However, in the `ELBERT` framework, the constraint term $\max_{g \in G} \frac{\eta_g^S(\pi)}{\eta_g^D(\pi)} - \min_{g \in G} \frac{\eta_g^S(\pi)}{\eta_g^D(\pi)}$ does not have a Bellman equation. Although both of $\eta_g^S(\pi), \eta_g^D(\pi)$ have Bellman equation since they are in the form of cumulative sum, it was previously unclear how to estimate the policy gradient of their ratio $\frac{\eta_g^S(\pi)}{\eta_g^D(\pi)}$. To adress this, in Section 3.2 we propose the `ELBERT` Policy Optimization (`ELBERT`-PO) framework that analytically derives the policy gradient of the constraint term.

**Step-wise safety constraints**   The second category is learning a policy that transits over only "safe" states, where the risk is less than a threshold $\delta$ at every timestep (Wachi & Sui, 2020). Mathematically, the goal is formulated as

$$\max_{\pi} \eta(\pi) \quad \text{s.t.} \quad C(s_t) \leq \delta, \ \forall t, \tag{18}$$

where $C : \mathcal{S} \to \mathbb{R}$ is the risk function. This constrained RL framework has step-wise constraints, which is different from `ELBERT` where the fairness constraint $\max_{g \in G} \frac{\eta_g^S(\pi)}{\eta_g^D(\pi)} - \min_{g \in G} \frac{\eta_g^S(\pi)}{\eta_g^D(\pi)} \leq \delta$ considers all future time steps. Therefore, techniques for this category of constrained RL cannot be directly applied in the `ELBERT` framework.

## D  EXPERIMENTAL DETAILS

### D.1  FULL DESCRIPTION OF THE ENVIRONMENTS

**Lending**   We consider the case of credit approval for lending in a sequential setting. As the agent in this scenario, a bank decides whether to accept or reject loan requests from a stream of applicants who arrive one-by one in a sequential manner. At each time $t$, an applicant from one of the two

groups arrives. More specifically, the applicant's group ID $g_t$ is sampled uniformly from $G = \{1, 2\}$. Given the current applicant's group ID $g_t \in \{0, 1\}$, the corresponding credit score $c_t \in \{1, 2, \cdots, C\}$ is sampled from the credit distribution $\boldsymbol{\mu}_{t,g_t} \in \Delta(C)$, where $\Delta(C)$ denotes the set of all discrete distributions over $\{1, 2, \cdots, C\}$. We note here that the credit score distributions of both groups, $\boldsymbol{\mu}_{t,1}$ and $\boldsymbol{\mu}_{t,2}$ are time-varying and will introduce their dynamics in detail later. Regardless of their group IDs $g_t$, the applicants with higher credit score is more likely to repay (i.e., $Y_t = 1$), whether the loan is approved (i.e., $\hat{Y}_t = 1$) or not (i.e., $\hat{Y}_t = 0$). Group 2 is disadvantaged with a lower mean of initial credit score compared to Group 1 at the beginning of the sequential decision-making process. The agent makes the decision $\hat{Y}_t \in \{0, 1\}$ using the observation $g_t$ and $c_t$. With $\hat{Y}_t$ and $Y_t$, the agent gets an immediate reward $R_t$ (agent's earned cash at step $t$), and the credit score distribution of the group $g_t$ changes depending on $\hat{Y}_t$ and $Y_t$. Specifically, the credit score of the current applicant shifts from $c_t$ to a new score $c'_t$, leading to the change of the credit score distribution of group $g_t$ as follows, where the constant $\epsilon$ is the dynamic rate.

$$\boldsymbol{\mu}_{t+1,g_t}(c'_t) - \boldsymbol{\mu}_{t,g_t}(c'_t) = \boldsymbol{\mu}_{t,g_t}(c_t) - \boldsymbol{\mu}_{t+1,g_t}(c_t) = \varepsilon \geq 0. \tag{19}$$

The fairness criterion is the long-term extension of Equal Opportunity and the group well-being is measured by the true positive rate. Specifically, the bias of a policy is defined as follows.

$$\left| \frac{\sum_t \mathbb{1}\{G_t = 0, Y_t = \hat{Y}_t = 1\}}{\sum \mathbb{1}\{G_t = 0, Y_t = 1\}} - \frac{\sum_t \mathbb{1}\{G_t = 1, Y_t = \hat{Y}_t = 1\}}{\sum \mathbb{1}\{G_t = 1, Y_t = 1\}} \right| \tag{20}$$

**Infectious disease control: original version.** In this environment, the agent is tasked with vaccinating individuals within a social network to minimize the spread of a disease (Atwood et al., 2019). We first introduce the original set up used in Yu et al. (2022) and in the next paragraph, we modify the environment to become more challenging. In this environment, individuals from two groups $G = \{1, 2\}$ are formulated as the nodes $v \in V$ in a social network $N$ connected with the edges $E$. Every individual has a health state from $\{H_S, H_I, H_R\}$ for being susceptible, infected and recovered. The state space of the RL agent is characterized by the health states of all individuals, i.e. $S = \{H_S, H_I, H_R\}^{|V|}$. A random individual in $N$ gets infected at the beginning of an episode. At each time step, the agent chooses one individual or no one to vaccinate and therefore the action space is the set of all individuals and the null set $V \cup \emptyset$. As for the dynamics, without vaccination, a susceptible individual gets infectious with probability that depends on the number of infectious neighbors. Specifically, without the vaccine, a susceptible individual $v$ will get infected with probability of $1 - (1 - \tau)^{I_N(v,H)}$, where $0 < \tau \leq 1$ and $I_N(v, H)$ is the number of infected individuals that are connected to the individual $v$. $\tau = 0.1$ is used. For those individuals in the susceptible state and receiving an vaccine, they will directly transit to the recovery state. A infected individual will get recovered with probability $\rho = 0.005$ without vaccination, and stay infected if vaccinated. The immediate reward is the percentage of health individuals (including being susceptible and recovered) in the whole network at the current step.

The fairness criterion is defined as

$$\left| \frac{\sum_t \text{vaccinations given}_{1t}}{\sum_t \text{newly infected}_{1t}} - \frac{\sum_t \text{vaccinations given}_{2t}}{\sum_t \text{newly infected}_{2t}} \right| \tag{21}$$

where vaccinations given$_{gt}$ and newly infected$_{gt}$ are the number of vaccinations given to individuals from group $g$ and the number of newly infected individuals from group $g$ at time $t$.

**Infectious disease control: harder version.** In the original setting used in Yu et al. (2022), the recovery state is absorbing: the individual in the recovery state will not be susceptible or infected again. To make the environment more challenging, we modify the environment so that the recovered individuals will become susceptible again with probability $\mu = 0.2$. This modification inject more stochasticity into the environment and makes learning more challenging. Other parameters are kept the same as the original settings. Note that the results in Section 5 is on this harder environment.

**Attention allocation: original version.** In the original version of this environment used in Yu et al. (2022), the agent's task is to allocate 6 attention units to 5 sites (groups) to discover incidents, where each site has a different initial incident rate. The agent's action is $a_t = \{a_{g,t}\}_{g=1}^5$,

where $a_{g,t}$ is the number of allocated attention units for group $g$. The number of incidents $y_{g,t}$ is sampled from $\text{Poisson}(\mu_{g,t})$ with incident rate $\mu_{g,t}$ and the number of discovered incident is $\hat{y}_{g,t} = \min(a_{g,t}, y_{g,t})$. The incident rate changes according to $\mu_{g,t+1} = \mu_{g,t} - d \cdot a_{g,t}$ if $a_{g,t} > 0$ and $\mu_{g,t+1} = \mu_{g,t} + d$ otherwise, where the dynamic rate $d$ is a constant. The agent's reward is $R(s_t, a_t) = \sum_g \hat{y}_{g,t} - \zeta \sum_g (y_{g,t} - \hat{y}_{g,t})$, where the coefficient $\zeta$ balances between the discovered and missed incidents. In the original version, $\zeta = 0.25$ and $d = 0.1$. The initial incident rates are given by

$$\{\mu_{g,0}\}_{g=1}^5 = \{8,\ 6,\ 4,\ 3,\ 1.5\}. \tag{22}$$

The group well-being is defined as the ratio between the total number of discovered incidents over time and the total number of incidents, and thus the bias is defined as

$$\max_{g \in G} \frac{\sum_t \hat{y}_{g,t}}{\sum_t y_{g,t}} - \min_{g \in G} \frac{\sum_t \hat{y}_{g,t}}{\sum_t y_{g,t}}. \tag{23}$$

**Attention allocation: harder version.** To modify the environment to be more challenging, we consider a more general environment by introducing more complexity. Different from the original setting in Yu et al. (2022) where the dynamic rate is the same among groups, we consider a more general case where the dynamic rates vary among different groups. Moreover, for the group $g$, the dynamic rate for increasing incident rate $\overline{d}_g$ is different from that for decreasing incident rate $\underline{d}_g$. Specifically, the incident rate changes according to $\mu_{g,t+1} = \mu_{g,t} - \underline{d}_g \cdot a_{g,t}$ if $a_{g,t} > 0$ and $\mu_{g,t+1} = \mu_{g,t} + \overline{d}_g$ otherwise, where the constants $\underline{d}_g$ and $\overline{d}_g$ are the dynamic rates for reduction and growth of the incident rate of group $g$. The parameters are given by the following.

$$\{\underline{d}_g\}_{g=1}^5 = \{0.004,\ 0.01,\ 0.016,\ 0.02,\ 0.04\},\ \{\overline{d}_g\}_{g=1}^5 = \{0.08,\ 0.2,\ 0.4,\ 0.8,\ 2\} \tag{24}$$

Meanwhile, we increase the number of attention units to allocate from 6 to 30 to expand the action space for more difficulty and modify the initial incident rates to

$$\{\mu_{g,0}\}_{g=1}^5 = \{30,\ 25,\ 22.5,\ 17.5,\ 12.5\}. \tag{25}$$

The agent's reward is $R(s_t, a_t) = -\zeta \sum_g (y_{g,t} - \hat{y}_{g,t})$, i.e., the opposite of the sum of missed incidents. Here $\zeta = 0.25$. Note that the reward function in this harder version is different from the original setting.

**Explanation of the harder environment.** The new version of the attention environment is more challenging for learning a fair policy with high rewards due to the following reasons. **(1)** The higher number of attention units indicates the larger action space in which searching for the optimal policy will be more challenging. **(2)** For all groups, the increasing dynamic rates are much higher than the decreasing dynamic rates, making it harder for the incident rate to decrease. **(3)** The disadvantaged groups, i.e., the groups with higher initial incident rates, have lower dynamic rates for both decreasing and increasing incident rate. This makes learning a fair policy harder since lower decreasing dynamic rates make the incident rates harder to decrease, and lower increasing dynamic rates means the policy could allocate less units to these groups without harming the reward too much, causing increasing bias. Note that the experiment in the attention environment in Section 5 uses this harder environment.

**Summary of all environments** (1) Lending. (2) Original Attention. (3) Harder Attention. (4) Original Infectious. (5) Harder Infectious. Note that the results in Section 5 are on (1), (3) and (5). Results of other environments are in Appendix D.3.

## D.2 Hyperparameters

For the learning rate, we use $10^{-6}$ in the original attention allocation environment and $10^{-5}$ in other four environments. We train for $2 \times 10^6$ time steps in the lending environment, $10^7$ time steps in the original infectious disease control environment, $2 \times 10^7$ time steps in the original attention allocation environment, and $5 \times 10^6$ time steps in two harder environments (infectious disease control and attention allocation).

Before listing the hyperparameters for baselines, we first briefly introduce two baselines R-PPO and A-PPO used in Yu et al. (2022). In Yu et al. (2022), it is assumed that there exists a fairness measure

function $\Delta$ so that $\Delta(s)$ measures the bias of the state $s$. In practice, $\Delta(s_t)$ is computed using the bias up to time $t$, which depends on the previous state action pairs. R-PPO directly modifies the reward function so that a larger bias results in smaller immediate reward. Specifically, R-PPO modifies the reward function into

$$R^{\text{R-PPO}}(s_t, a_t) = R(s_t, a_t) + \zeta_1 \Delta(s_t). \tag{26}$$

A-PPO modifies the advantage function to encourage the bias at the next time step to be smaller than the current step. Specifically, it modifies the standard advantage function $\hat{A}(s_t, a_t)$ into

$$\hat{A}^{\text{A-PPO}}(s_t, a_t) = \hat{A}(s_t, a_t) + \beta_1 \min(0, -\Delta(s_t) + \omega) + \beta_2 \begin{cases} \min(0, \Delta(s_t) - \Delta(s_{t+1})) & \text{if } \Delta(s_t) > \omega \\ 0 & \text{otherwise} \end{cases} \tag{27}$$

The hyperparameters for R-PPO and A-PPO in each environment are shown in Table 1.

| Environments | | ELBERT-PO | R-PPO | A-PPO |
|---|---|---|---|---|
| Lending | | $\alpha = 2 \times 10^5$ | $\zeta_1 = 2$ | $\beta_1 = \beta_2 = 0.25$ $\omega = 0.005$ |
| Infectious disease control | Original | $\alpha = 10$ | $\zeta_1 = 0.1$ | $\beta_1 = \beta_2 = 0.1$ $\omega = 0.05$ |
| | Harder | $\alpha = 50$ | | |
| Attention allocation | Original | $\alpha = 50$ $\beta = 20$ | $\zeta_1 = 10$ | $\beta_1 = \beta_2 = 0.15$ $\omega = 0.05$ |
| | Harder | $\alpha = 2 \times 10^4$ $\beta = 20$ | $\zeta_1 = 20$ | |

Table 1: Hyperparameters of ELBERT-PO and two baseline methods (R-PPO and A-PPO).

All experiments are run on NVIDIA GeForce RTX 2080 Ti GPU.

### D.3 MORE EXPERIMENTAL RESULTS

**Results on original attention and infectious disease control environments.** The performance of ELBERT-PO and baselines in the original versions of the attention and infectious disease control environments are shown in Figure 5. **(1) Infectious (original).** ELBERT-PO achieves the highest reward among all baselines, including G-PPO. As for the bias, ELBERT-PO, R-PPO and A-PPO obtains almost the same low bias. This suggests that the original infectious disease control environment is not challenging enough to distinguish the bias mitigation ability between ELBERT-PO and the baselines. **(2) Attention (original).** In this environment, we find that G-PPO, without any fairness consideration, achieves very low bias (around 0.05). This indicate that the original attention environment is too easy in terms of bias mitigation, and thus it must be modified to be more challenging. All methods obtain almost the same low bias. The results on the original version of both environments motivate us to modify them to be more challenging. The experiments in Section 5 shows that ELBERT-PO has huge advantage in bias mitigation on more challenging environments.

**Ablation on the bias coefficient $\alpha$.** The learning curve of rewards and bias from ELBERT-PO with various values of the bias coefficient $\alpha$ in different environments are shown in Figure 6. **(1)** Rewards. In most cases, increasing $\alpha$ leads to slower convergence in rewards. However, increasing $\alpha$ can either increase the reward (lending), decrease the reward (original and harder infectious) or be irrelevant to reward (original attention allocation). This shows that whether or not there is an intrinsic trade-off between reward and bias depends on the environment. **(2)** Biases. Using a relatively $\alpha$ leads to lower bias. However, when $\alpha$ is too large, the bias might increase again. This may be due to the instability of the bias term when $\alpha$ is too large.

**Ablation on the temperature $\beta$ of the soft bias.** The learning curves of rewards and bias with various values of temperature $\beta$ of the soft bias in the harder version of attention allocation environment

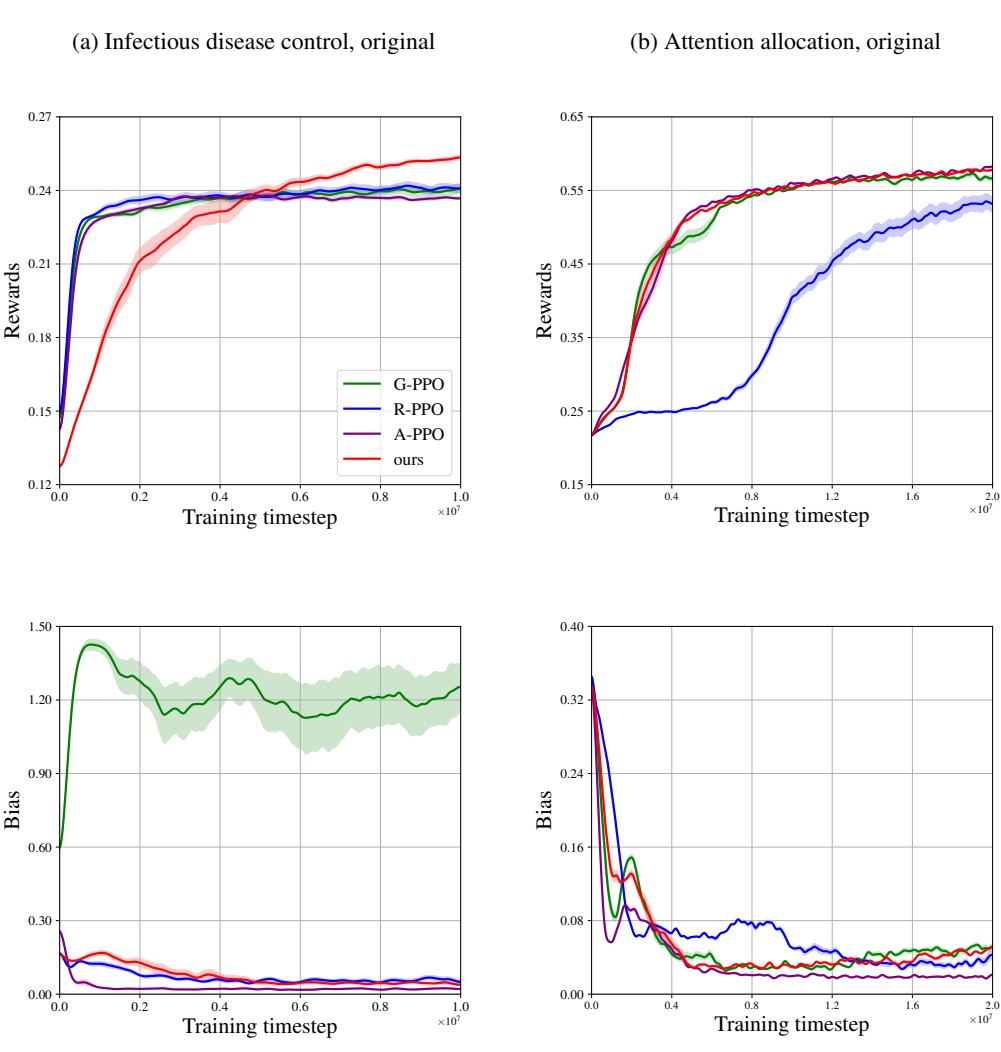

Figure 5: Rewards and bias of `ELBERT`-PO (ours) and three other RL baselines (A-PPO, G-PPO, and R-PPO) in two original environments (infectious disease control and attention allocation) from Yu et al. (2022).

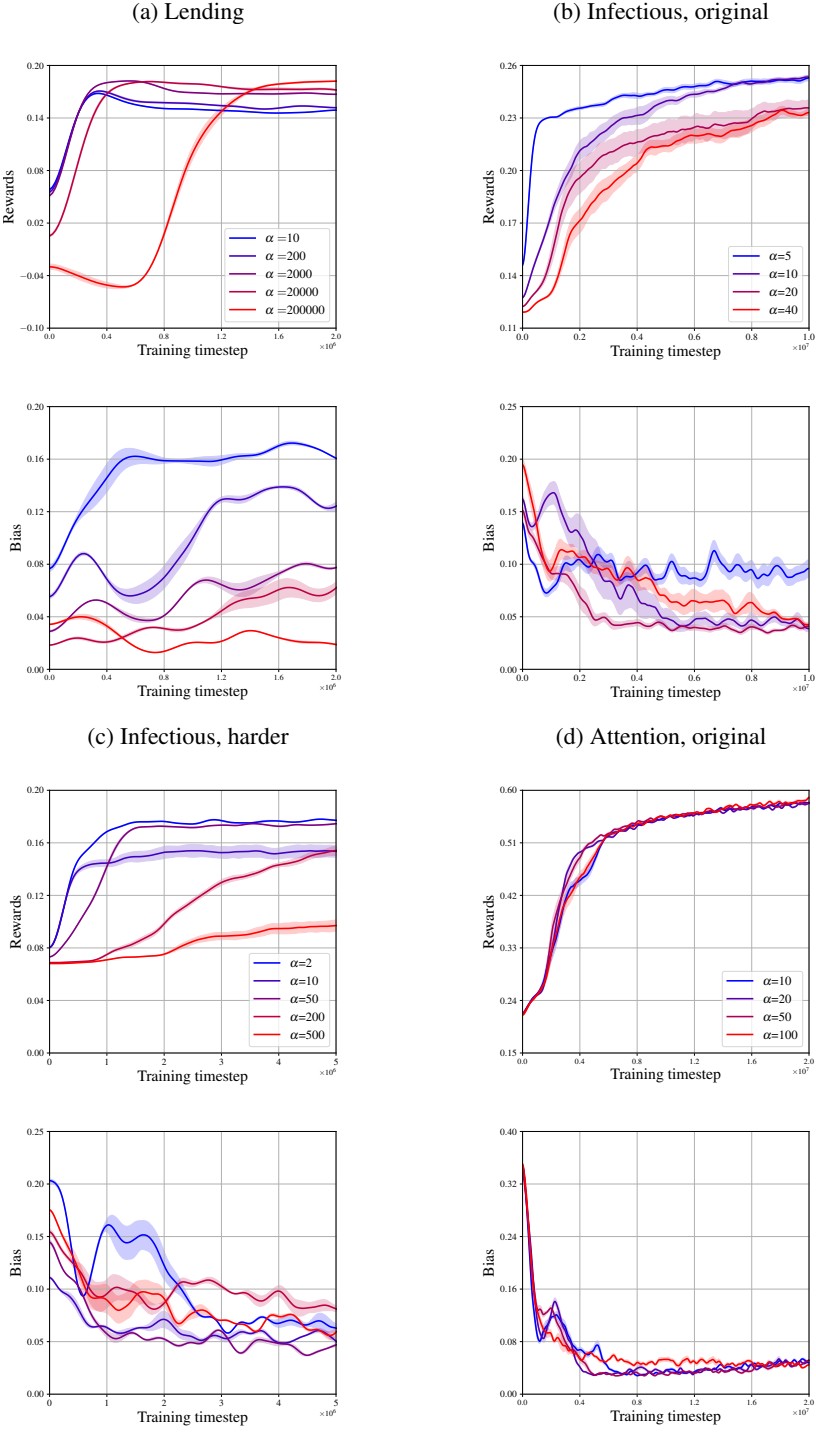

Figure 6: Learning curve of ELBERT-PO on four environments (lending, original and harder infectious disease control, and original attention allocation) with different $\alpha$. The learning curve on the harder attention allocation environment is shown in Figure 4.

are shown in Figure 7. We observe that **(1)** when $\beta$ is very small ($\beta = 1$), the bias is relatively large. This is because as shown in Proposition 3.3, the gap between the soft bias and bias is larger when $\beta$ is smaller, and therefore minimizing the soft bias may not be effective in minimizing the true bias. **(2)** When $\beta$ is very large ($\beta = 100$), at the beginning of training, the bias decreases slightly slower than when $\beta$ is moderate ($\beta = 20$). Also, the reward is observed to be less stable when $\beta = 100$. This is probably due to the non-smoothness of optimization, since when $\beta$ is very large, the soft bias is very close to the bias, which is non-smooth and only depends on the maximal and minimal group benefit rate. **(3)** Despite of the slight difference in convergence rate, the bias converges to the same value for $\beta = 20$ and $\beta = 100$. This indicates that ELBERT-PO is not sensitive to $\beta$, provided that $\beta$ is reasonably large.

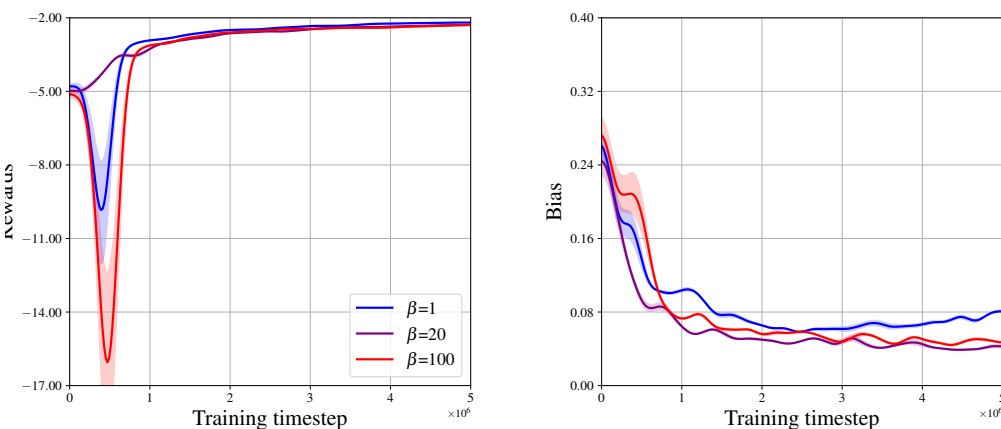

Figure 7: Learning curve of ELBERT-PO on the harder version of the attention allocation environment with different $\beta$.

**Supply and demand of each group over training time steps.** To explore how ELBERT-PO minimizes the bias, we examine how the supply and demand of the advantaged and disadvantaged groups vary over training steps. In the multi-group setting, at each training step we consider the most advantaged and disadvantaged group. The corresponding results of ELBERT-PO and the other three baseline methods in three main environments (lending, harder infectious, and harder attention), are shown in Figure 8. **(1) Lending.** The bias is reduced mainly by both decreasing demand and increasing supply of the disadvantaged group. When using ELBERT-PO, in addition to reducing the bias, the group benefit rates of both groups actually increase (but their difference decreases). **(2) Infectious (harder).** The demand of both groups seems not influenced much by decision-making system. The bias is reduced mainly by increasing the supply of the disadvantaged group and decreasing supply of the advantaged group. **(3) Attention (harder).** In this case, the bias is reduced largely by increasing the demand of the advantaged group and decreasing the demand of the disadvantaged group. To sum up, the bias is reduced in different ways on different environments.

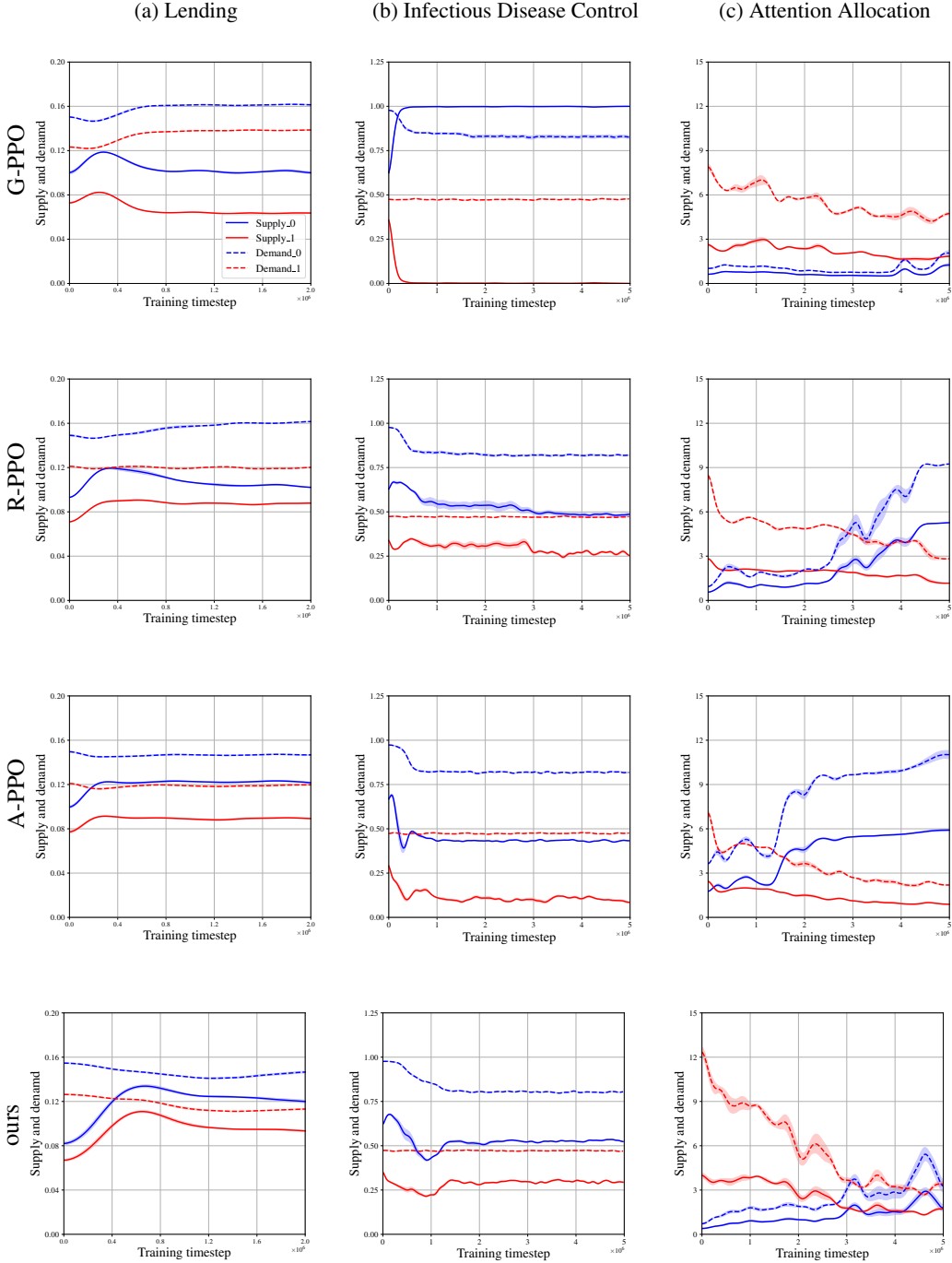

Figure 8: Group demands and supplies of advantaged group 0 and disadvantaged group 1 with ELBERT-PO (ours) and three other RL baselines (A-PPO, G-PPO, and R-PPO) in three environments (lending, harder infectious disease control and harder attention allocation). Each row shows the results of one method and each column shows the results on one environment.

# E   ADDRESSING POTENTIAL LIMITATION

**Demand in the static and long-term settings**   In the static setting, static fairness notions (such as Demographic Parity and Equal Opportunity) only consider the supply-demand ratio and do not explicitly consider the absolute value of demand itself. This is because the demand is typically not considered as controllable by the decision maker in the static setting. Our ELBERT framework adapts static fairness notions to long-term settings, and consider the ratio between cumulative supply and demand. The ELBERT does not explicitly consider the absolute value of demand either. Note that in the long-term setting, demand in the future time steps can be affected by the decision maker.

**Overall potential limitation**   One limitation of ELBERT is that it focuses on long-term adaptations of static fairness notions, which mainly consider the supply-demand ratio but not the absolute value of demand itself. However, in real world applications, extra constraints on demand might be needed. For example, the demand should not be too large (e.g. when demand is the number of infected individuals in the infectious environment) or too small (e.g. when demand is the number of qualified applicants in the lending environment).

Specifically, in ELBERT framework and in all the metrics used by prior works (Yu et al., 2022; D'Amour et al., 2020; Atwood et al., 2019), reducing the bias typically encourages the Long-term Benefit Rate of the disadvantaged group to increase, which can decrease its group demand. Although decreasing the demand of the disadvantaged group is beneficial in some cases (e.g. decreasing the number of infected individual in the infectious environment), overly decreasing demand can be problematic in other cases (e.g. when demand is the number of qualified applicants in the lending environment) due to real world consideration.

**Solution**   In the following, we show that ELBERT-PO still works when we incorporate additional terms to regularize demand in the objective function. For illustration, assume that we would like the demand of the disadvantaged group (group 1) to be not too small (e.g. in the lending environment). Therefore, we can add a regularization term $\eta_1^D(\pi)$ to the objective function to maximize:

$$J^{\text{reg}}(\pi) = \eta(\pi) - \alpha b(\pi)^2 + \zeta \eta_1^D(\pi) = \eta(\pi) - \alpha \left( \frac{\eta_1^S(\pi)}{\eta_1^D(\pi)} - \frac{\eta_2^S(\pi)}{\eta_2^D(\pi)} \right)^2 + \zeta \eta_1^D(\pi), \quad (28)$$

where $\zeta$ controls the regularization strength for keeping the demand of group 1 from being too small.

As in Section 3.2, we need to compute its policy gradient in order to use standard policy optimization algorithms like PPO. Note that this is easy since the extra regularization term $\eta_1^D(\pi)$ is in the form of a standard cumulative reward with known policy gradient formula. Therefore, combining with the policy gradient formula without the regularization term in Equation (4), we have that:

$$\nabla_\pi J^{\text{reg}}(\pi) = \mathbb{E}_\pi \left\{ \nabla_\pi \log \pi(a_t|s_t) \left[ A_t - \alpha \sum_{g \in G} \frac{\partial h}{\partial z_g} \left( \frac{1}{\eta_g^D(\pi)} A_{g,t}^S - \frac{\eta_g^S(\pi)}{\eta_g^D(\pi)^2} A_{g,t}^D \right) + \zeta A_{1,t}^D \right] \right\} \quad (29)$$

Therefore, we only need to use the demand-regularized version of the fairness-aware advantage function $A_t^{\text{fair Reg}} = A_t - \alpha \sum_{g \in G} \frac{\partial h}{\partial z_g} \left( \frac{1}{\eta_g^D(\pi)} A_{g,t}^S - \frac{\eta_g^S(\pi)}{\eta_g^D(\pi)^2} A_{g,t}^D \right) + \zeta A_{1,t}^D$ and apply Algorithm 1.

**Demand in our experiments**   In Figure 8 in Appendix D.3, we visualize how demand and supply changes during training for all methods in the three environments. Note that in all algorithms the demand is not regularized. We did not notice any aforementioned problematic and dramatic change in demand of the disadvantaged group. Specifically, *(1) Lending.* The demand (number of qualified applicants) of the disadvantaged group only mildly decreases using methods with fairness considerations (ELBERT-PO, A-PPO and R-PPO). The supply of the disadvantaged group increases a lot. *(2) Infectious.* The demand of the disadvantaged group is barely affected by all algorithms. *(3) Attention.* The demand (number of incidents) of the disadvantaged group goes down when using methods with fairness considerations. Although the demand regularization technique above is not needed in these environments, it might be crucial in other applications.

