# OpenReview forum: "Equal Long-term Benefit Rate: Adapting Static Fairness Notions to Sequential Decision Making"
_ICLR.cc/2024/Conference — Submitted to ICLR 2024_

### Official Review · Reviewer_aJ3j · 2023-10-31

**Soundness:** 3 good
**Presentation:** 3 good
**Contribution:** 1 poor
**Rating:** 5
**Confidence:** 3

**Summary:**

This paper introduces a new long-term fairness notion called ELBERT that considers varying temporal importance and adapts static fairness principles to sequential settings. The previous long-term fairness notion in the MDP setting, which primarily considers the sum of step bias, could potentially lead to a false sense of fairness. Later, the paper relaxes the objective (RL with ELBERT, equation 1) into a unconstrained optimization problem (equation 2), and shows the policy gradient of the Long-term benefit rate can be transformed into a traditional policy gradient and uses a standard optimization method to solve for the gradient. Experimental results demonstrate the efficiency of the proposed method in terms of reducing bias.

**Strengths:**

The paper is well-written and relatively easy to follow. The new proposed fairness notion is intuitive, and the demonstration of Figure 1 emphasizes the importance of considering variation in temporal importance. The authors provide propositions to help understand their proposed method and clearly state their proposed algorithm.

**Weaknesses:**

The major concern is that the contribution of the paper feels relatively marginal besides the major contribution is the newly proposed long-term fairness notion. In addition, The proposed method involves a lot of relaxations/approximations, without enough rigorous theoretical justification, as evident from the questions raised below. Consequently, it appears to lean more towards a heuristic application of conventional MDP techniques rather than introducing a genuinely innovative concept. I would consider increasing my score if the authors address these concerns.

**Questions:**

1. How does solving (2) equivalent to solving (1)? I assume it's related to the Lagrangian form? Since equation (1) is the primary objective function of the paper, the authors should consider adding a detailed justification of the equivalency between the two equations.

2. What is the "Bellman Equation"? It might be unclear to a non-expert in MDP, but it seems like an important concept.

3. Continuing the above question, it's unclear how proposition 3.1 helps overcome the difficulty of being unable to compute $b(\pi)$ directly.

4. The authors demonstrate in Figure 1 when ELBERT is strictly better than the sum of step-wise bias. Does it imply that ELBERT is always strictly better than the traditional fairness notion?

---

> ### Author Response · Authors · 2023-11-14
> **Our contributions include solving an fairness objective which was not studied in RL literature**
>
> We thank Reviewer aJ3j for the detailed and insightful feedback. We are encouraged that the reviewer finds our fairness notion intuitive and acknowledges the importance of variation in temproal importance. Below we address the reviewer's concerns in detail.
>
> ---
> > W1.1: The major concern is that the contribution of the paper feels relatively marginal besides the major contribution is the newly proposed long-term fairness notion.
>
> It seems that our technical contributions of the bias mitigation method, ELBERT-PO, are not fully clear to the reviewer. We clarify below.
>
> 1. Our technical framework relies on Reinforcement Learning (RL). However, the mathematical form of our proposed bias metric was not studied by RL. This is because RL generally studies how to optimize the cumulative reward, which is very distinct from our proposed bias metric.
> 2. In order to decrease bias, we need to compute the policy gradient of the bias, which was not obvious from RL literature. We have to mathematically derive the policy gradient of bias and figure out how to integrate it with popular RL methods. This should be regarded as one of our major contributions.
>
>
> > W1.2: In addition, The proposed method involves a lot of relaxations/approximations, without enough rigorous theoretical justification, as evident from the questions raised below. Consequently, it appears to lean more towards a heuristic application of conventional MDP techniques rather than introducing a genuinely innovative concept
>
> It is incorrect that our ELBERT-PO method involves `"a lot of"` approximations without `"enough rigorous theoretical justification"`.
>
> Our method only has two approximations:
> 1. It uses a regularization-based objective (Equation 2) to solve a hard-constrained objective (Equation 1). This is indeed an approximation and is also a very common practice, which we clarify in Q1.
> 2. In multi-group settings, we propose to use the "soft bias" instead of bias to avoid non-differentiability issue, which is theoretically justified by Proposition 3.3. This is also common and important in machine learning, when the evaluation metrics in non-differentiable but you need a differentiable training objective. For example, in tradition classification problem we use accuracy for evaluation and use cross-entropy, the differentiable surrogate of accuracy, for training.

---

> ### Author Response · Authors · 2023-11-14
> **Clarifying the challenge of bias gradient and how it is solved**
>
> ___
> > Q1: How does solving (2) equivalent to solving (1)? I assume it's related to the Lagrangian form? Since equation (1) is the primary objective function of the paper, the authors should consider adding a detailed justification of the equivalency between the two equations.
>
> Here we adopt the penalty method [R1], using a regularization-based objective Eq (2) to address the hard-constrained problem Eq (1). This is because in practice the hard-constrained problem can be too restrictive and might lead to infeasibility in the solution space (e.g. no policy is under the bias threshold $\epsilon$). Also, using a unconstrained regularization-based objective facilitates the gradient-based optimization.
>
> In the regularization-based objective Eq (2), the relative importance of reward and bias is controlled by the bias coefficient $\alpha$. Indeed, we observe in the experiment that increasing $\alpha$ typically reduces the bias. Our method achieves the lowest bias among all baseline and is thus very effective.
>
> We also would like to mention that although unconstrained, the objective in Eq (2) is not easy to solve: its gradient is not directly given by traditional Reinforcement Learning. It is our contribution to compute its gradient in Eq (4).
>
>
> Ref:
> [R1] Nocedal, Jorge, and Stephen J. Wright, eds. Numerical optimization. New York, NY: Springer New York, 1999.
>
> ---
> > Q2: What is the "Bellman Equation"? It might be unclear to a non-expert in MDP, but it seems like an important concept.
>
> Bellman Equation is a foundational equation for Reinforcement Learning. We have added appropriate reference in Section 3.1. The following is a brief summary of Bellman Equation and we refer the reviewer to [R2] for further details.
>
> In Reinforcement Learning (RL) problems, it is important to consider the value functions. For example, the state value function in the following characterizes the cumulative reward of a certain state following a policy $\pi$.
> $$V^\pi(s)=\mathbb{E}\_\pi\left[\sum\_{t=0}^\infty\gamma^t R(s\_t,a\_t)\Big|\pi,s\_0=s\right].$$
>
> It turns out value functions satisfy the Bellman Equation
>
> $$V^\pi(s)=\sum_a\pi(a|s)\sum_{s'}T(s'|s,a)[R(s,a)+\gamma V^\pi(s')]$$
>
> Commonly used advantaged-based policy optimization methods (e.g. PPO [R3]) rely on Bellman Equation in order to compute the policy gradient of the cumulative reward.
>
> Ref
> [R2] Sutton, Richard S., and Andrew G. Barto. Reinforcement learning: An introduction. MIT press, 2018.
> [R3] Schulman, John, et al. "Proximal policy optimization algorithms." arXiv preprint arXiv:1707.06347 (2017).
>
> ---
> > Q3: Continuing the above question, it's unclear how proposition 3.1 helps overcome the difficulty of being unable to compute $b(\pi)$ directly.
>
> It was previously unclear how to compute the gradient of $b(\pi)$ from RL literature. Proposition 3.1 shows how to compute the gradient of $b(\pi)$ (and $J(\pi)$) using $\nabla_{\pi}\eta_{g}^D$ and $\nabla_{\pi}\eta_{g}^S$, which we do know how to compute from RL literature. We provide more details below.
>
> 1. It was unclear how to directly compute the gradient of $b(\pi)=\left(\frac{\eta_1^S(\pi)}{\eta_1^D(\pi)}-\frac{\eta_2^S(\pi)}{\eta_2^D(\pi)}\right)^2$ since it is not in the form of cumulative sum (e.g. $\sum_{t=0}^\infty\gamma^t R(s_t,a_t)$). We only know how to compute gradient of cumulative sum from RL literature.
> 2. $\eta_{g}^D$ and $\eta_{g}^S$ are indeed in the form of cumulative sum (see Definition 2.2) so we know how to compute $\nabla_{\pi}\eta_{g}^D$ and $\nabla_{\pi}\eta_{g}^S$.
> 3. Proposition 3.1 computes the previously unknown $b(\pi)$ using $\nabla_{\pi}\eta_{g}^D$ and $\nabla_{\pi}\eta_{g}^S$ that are known in RL literature.
>
> ---
> > Q4: The authors demonstrate in Figure 1 when ELBERT is strictly better than the sum of step-wise bias. Does it imply that ELBERT is always strictly better than the traditional fairness notion?
>
> We assume the `"traditional fairness notion"` mentioned by the reviewer means fairness notions in static settings, such as Demographic Parity.
>
> Our ELBERT framework **adapts** these fairness notions in static settings to sequential decision making settings, since these static notions cannot be directly used in sequential settings. We should not compare the two, since static fairness notions only work in static settings, where our notion works in sequential settings.
>
> ---
>
> Thank you again for your time and effort in reviewing our paper! Please let us know if the above explanations do not address your concerns. We are happy to answer any further questions.

---

> ### Author Response · Authors · 2023-11-19
> **Does our response address your concerns?**
>
> Dear reviewer aJ3j,
>
> As the stage of the review discussion is ending soon, we would like to kindly ask you to review our revised paper as well as our response and consider making adjustments to the scores. Please let us know if there are any other questions. We would appreciate the opportunity to engage further if needed.
>
> Best regards,
>
> Paper1977 Authors

---

> > ### Author Response · Authors · 2023-11-21
> > **Does our response address your concerns?  Edit**
> >
> > Dear reviewer aJ3j,
> >
> > As the stage of the review discussion is ending soon, we would like to kindly ask you to review our revised paper as well as our response and consider making adjustments to the scores. Please let us know if there are any other questions. We would appreciate the opportunity to engage further if needed.
> >
> > Best regards,
> >
> > Paper1977 Authors

---

> > > ### Comment · Reviewer_aJ3j · 2023-11-21
> > > **Thank you for your response**
> > >
> > > I've carefully read the response from the authors, and my concerns have been addressed.
> > >
> > > I encourage the authors to incorporate the additional comments to the main paper to 1) better emphasize its contribution and 2) make it more accessible to readers who are not that familiar with techniques from RL literature.

---

> > > > ### Author Response · Authors · 2023-11-21
> > > > **Thank you for your follow-up comment**
> > > >
> > > > Thank you for your response! We are glad that all the concerns have been addressed.
> > > >
> > > > As for paper writing, we have included additional explanations and reference in our revised manuscript about RL and the challenge our method solves. For the latest version:
> > > >
> > > > 1. In the introduction, we have summarized the challenge as ``optimzing Long-term Benefit Rate is challenging since it is not in the standard form of cumulative reward in RL and how to compute its policy gradient was previously unclear.`` and our solution as ``To address this, we show that its policy gradient can be analytically reduced to the standard policy gradient in RL by deriving the fairness-aware advantage function``.
> > > > 2. In the methodology section 3.1, we have dedicated the second paragraph (``Challenge: policy gradient of $b(\pi)$``) to explain why the bias policy gradient is not studied by traditional RL. We have added the reference to Bellman equation and general RL knowledge.
> > > > 3. In section 3.2 we have laid out the mathematical roadmap to solve the aforementioned challenge of computing bias gradient.
> > > > 4. We dedicate the whole Appendix C to clarify in details what Bellman equation is, what objective traditional RL methods typically solve and why our objective cannot be directly solved by traditional RL.
> > > >
> > > > Since all the concerns have been addressed, we would like to kindly ask you to review our revised paper as well as our response and consider making adjustments to the scores.

---

### Official Review · Reviewer_1EzK · 2023-10-31

**Soundness:** 2 fair
**Presentation:** 3 good
**Contribution:** 2 fair
**Rating:** 3
**Confidence:** 4

**Summary:**

This paper introduces a novel long-term group fairness notion, where the (sensitive/demographic) group size is not determined at each time step such as in static fairness notion ("to decide fair relative to a group of individuals with certain demographics present at a certain time step, e.g., all females at one time step"), but the group is considered to be all individuals from a sensitive/demographic group across time steps ("to decide fair relative the cumulative group of individuals with certain demographics across all time steps, e.g., the entire group of females accumulated over all time steps"). The paper then introduces how to learn a policy adhering to their proposed fairness notion in a Markov Decision Making process setting.

**Strengths:**

[S1] The topic of fairness in sequential decision making, where individuals arrive on a rolling basis (at different time steps) and a decision has to be made at each time step, is an interesting one.

[S2] The proposed fairness notion is looking at group-based fairness notions, where the group is not defined by the amount of individuals to be decided on at a certain time step, but across time steps. This could be interesting in a hiring scenario, where a firm would hire each month (12 times a year) and finds enforcing demographic parity (DP, i.e., equal acceptance rates) difficult to fulfill at each month, but can commit to fulfilling DP on a yearly base (average across all hiring decisions of the year).

**Weaknesses:**

[W1] I have several difficulties to parse the example provided in the introduction. First, why is the proposed medical treatment a fair decision-making problem? Given the framing with individuals seeking medical care as "demand" and a medical facility providing healthcare as "supply", this appears to me a fair allocation/ranking problem, which is - to my undestanding - different from the general decision making, in having an upper bound on the (usually) positive decision. Second, what is the motivation for a decision-maker to aim for demographic parity (equal acceptance rates) in a medical treatment scenario, i.e., why is it the goal to provide the same proportion of service to blue and red patients? The motivation of the proposed fairness notion lives from the observation that it would be “unfair” to provide nobody of the group of 100 red people with a treatment at time step t, just because we do not provide the 1 blue person at time step t with treatment. I am struggeling to think of a real-world scenario, where a decision-maker would take such decisions, thus I have difficulties to follow the motivation for the example and thus the proposed fairness notion. As indnicated in [S2], it would be easier for me to see an application of this in a hiring scenario.

[W2] If I understood the example and fairness notion correctly, the demographic parity notion from the introduction is only expected to yield a different result than static one-step fairness notions, when i) demographic group sizes vary across time steps and ii) if a) decisions are trivial (i.e., either “accept all” or “accept nobody”) or b) we have one group with a group size 1. Here is a scenario, where i) is fulfilled, but ii) not: When we have 20 non-binary individuals and 100 binary individuals at t, then 80 non-binary individuals and 10 binary individuals at t+1 and fulfill DP with an acceptance rate p \in (0, 1) at each time step (such that DP-unfairness is zero), then the proposed long term notion yields the same as the single time step notion, e.g., p_t = 0.5, p_{t+1} = 0.25, then |(10+40)/(20+80) – (50+5) / (100+10)| = |50/100 – 55/110| = |0|. To my understanding from the example, the notion is only useful in a very specific case, which limits its practicality. I am happy to be corrected by the authors, if I misunderstood.

[W3] I find calling some time steps more “important” than others misleading. The authors call a time step “more important” for a demographic group, if there exist more of its members that are seeking a decision at that time step. This means, larger groups are paid more importance to than smaller groups. First, this is an ethical understanding that should be at least mentioned, if not discussed. Second, I find this understanding of fairness especially concerning, if the variance in the group sizes of individuals that arrive at each time step is biased or confounded by another factor. For a given demographic group (e.g., blue ones), individuals who tend to arrive in smaller groups may be treated unfairly in comparison to individuals, who tend to arrive in larger groups. This is because the fairness notion aggregates the blue individuals over all time steps to one large blue group and demands the cummulative decisions across all time steps to be fair to the cummulated blue groups. So this may yield to a type of inter-group unfairness.

[W4] The authors claim on page 2: “Without considering the variation in temporal importance within a group, these prior metrics lead to a false sense of fairness.” I strongly object the attempt to claim that prior fairness notions are “false”, which the authors do several times on page 2. Different fairness notions capture different understandings of fairness and which one to pick is very dependent on the context and scenario, and importantly should be at the heart of different research fields, such as philosophy, ethics, sociology etc. I advocate to remove these strong claims, and additionally merge the related work section with the introduction or put it right after it to contextualize the claims made as well as the motivate the need for a "new" fairness notion better.

[W5] In the introduction, the authors keep writing that they measure “bias”, but to my understanding, what they are measuring is not a (statistical, sampling, cofounding etc.) bias, but their moral/ethical understanding of “unfairness”.

[W6] In the conclusion the authors write “However, in real world applications, extra constraints on demand might be needed. For example, the demand should not be too large (e.g. when demand is the number of infected individuals) or too small (e.g. when demand is the number of qualified applicants).” I am failing to follow this argumentation. “Demand” in how the users understand it, is the number of individuals a decision-maker is deciding upon at each time step (e.g., loan applicants). I am failing to see, how and why these extra constraints might be needed.

[W7] There are missing citations for the background section, e.g., for Standard MDP. In the experimental section, there are missing citations for each of the three environments considered.


Minors
-	In times of increasing research on large language models, I would suggest the authors to pick a different name for their fairness notion than “ELBERT”, given the similarity to all the variations of BERT models.

**Questions:**

In addition to some of my questions / comments under "Weaknesses', I have the following questions:

[Q1] The authors write on page 3 “In many static fairness notions, the formulation of the group well-being can be unified as the ratio between supply and demand.“ I have difficulties to understand, where this understanding of demand and supply in fairness notions comes from. There is no citation given. To my understanding, at a classical sequential decision making scenario, there is at each time step a set of individuals that are required to be made a decision upon, and the decision maker takes the features of the individuals as input and decides usually aiming to maximize their utility while potentially adhering to some fairness notions. The concept of supply and demand comes from economics, where the general standard assumption is that goods demanded/supplied are uniform, i.e., every individual item is the same. This is not the case in fair decision making; every individual is different. There is not a demand and supply of credits and as long as a decision maker can supply credits they will. Instead, they will aim to give the credit to the individuals that they expect to have a high probability of paying back. Could the authors please clarify this?

---

> ### Author Response · Authors · 2023-11-14
> **Clarification of a foundamental misinterpretation of ELBERT**
>
> We thank Reviewer 1EzK for the detailed and insightful feedback. We are encouraged that the reviewer finds the topic of sequential decision making and our fairness notion interesting. Below we address the reviewer's concerns in detail.
>
> We will start with W3, which is a fundamental misinterpretation of our notion. Then we will respond to each weakness and question one-by-one.
>
>
> ---
> > W3: I find calling some time steps more “important” than others misleading. ... This means, larger groups are paid more importance to than smaller groups. First, this is an ethical understanding that should be at least mentioned, if not discussed. Second, I find this understanding of fairness especially concerning, if the variance in the group sizes of individuals that arrive at each time step is biased or confounded by another factor. For a given demographic group (e.g., blue ones), individuals who tend to arrive in smaller groups may be treated unfairly in comparison to individuals, who tend to arrive in larger groups. This is because the fairness notion aggregates the blue individuals over all time steps to one large blue group and demands the cummulative decisions across all time steps to be fair to the cummulated blue groups. So this may yield to a type of inter-group unfairness.
>
> This is a **fundamental misinterpretation** of our proposed notion. The reviewer claimed that in our proposed notion, `"larger groups are paid more importance to than smaller groups"` and `"individuals who tend to arrive in smaller groups may be treated unfairly in comparison to individuals, who tend to arrive in larger groups"`, which is incorrect.
>
> To clarify
> 1. Under our notion (first aggregating supply and demand over time, then computing the ratio), within a group, individuals from different timesteps have the **same** impact on bias computation. To see this, consider the red group in the motivating example: at time $t$ the group size is 100, and at $t+1$ the group size is 1. For any individual from either time $t$ or time $t+1$, the effect of the decision, changing from "rejection" to "acceptance" will change the Long-term Benefit Rate by $\frac{1}{101}$, which is the same for both $t$ and $t+1$.
> 2. Under prior notions (first computing step-wise ratio, then aggregating), within a group, individuals from timesteps with larger group size have smaller impact on bias computation, causing discrimination. To see this, consider the red group in the motivating example again: At time $t$, this group has 100 members, and at $t+1$, it reduces to just one member. Under these conditions, a decision shift from "rejection" to "acceptance" for an individual at time $t$ alters the step-wise ratio marginally, by $\frac{1}{100}$. In stark contrast, the same decision change for an individual at $t+1$ significantly affects the step-wise ratio, by $\frac{1}{1}$, which is considerably more impactful than the change at $t$. Consequently, when aiming to enhance the overall well-being of the red group, the individual at the smaller group size of 1 at $t+1$ becomes disproportionately influential, as decisions regarding this individual have a greater effect. This leads to a discriminatory outcome where individuals at time $t$, due to their minimal impact on fairness measures, are effectively marginalized.
>
> Therefore, prior notions do not equally treat individuals within the same group from different timesteps (with different number of people). The reviewer finds this concerning by saying `" I find this understanding of fairness especially concerning, if the variance in the group sizes of individuals that arrive at each time step is biased or confounded by another factor."` We are concerned too. **But this downside belongs to prior notions, not our proposed ELBERT**.
>
> Finally, it seems that there is confusion between the following two concepts: within a group:
> 1. When defining the bias, a timestep with higher group demand (number of individuals from that group) should not be treated the same as a timestep with lower group demand. This is indeed our claim. If one treats all timesteps the same, then individuals from different timesteps will not be treated equally.
> 2. An individual arriving at a timestep with larger number of people, will be treated as differently than the one from a timestep with lower number of people. As explained above, this is NOT true for our notion and is a downside of prior notions.

---

> ### Author Response · Authors · 2023-11-14
> **The motivating example involves both reward and fairness**
>
> > W1.1: ... the example provided in the introduction. First, why is the proposed medical treatment a fair decision-making problem? Given the framing with individuals seeking medical care as "demand" and a medical facility providing healthcare as "supply", this appears to me a fair allocation/ranking problem, which is - to my undestanding - different from the general decision making, in having an upper bound on the (usually) positive decision.
>
> The medical treatment allocation problem is a fair decision-making problem, where the decision maker maximizes the reward while ensuring fairness.
>
> 1. This example features as one of the experimental environments, detailed in Case 2, Section 5.1.
> 2. Excluding fairness considerations, this problem represents a typical sequential decision-making scenario where the objective is to maximize the reward, defined as the overall percentage of a healthy population over time.
> 3. With fairness in mind, the decision-maker aims to minimize bias, adhering to the principle of demographic parity. The notions of "group demand" and "group supply" are instrumental in bias measurement, viewed from individual perspectives, as fairness is evaluated at this level.
> 4. In our initial writing, we prioritized brevity and focused solely on fairness in the introduction, inadvertently omitting the reward aspect. This omission led to some confusion.
>
> To eliminate this ambiguity and reaffirm that our sequential decision-making framework encompasses both reward maximization and bias minimization, we have revised the introduction. It now features a Bank Lending example, which may be more relatable. The revisions include: (1) Defining the problem as a sequential decision-making process where the bank seeks to maximize long-term profits (the reward), and (2) acknowledging the bank's consideration of bias. (3) Additionally, the main figure in the introduction now explicitly represents both the reward and fairness-related elements.
>
>
> > W1.2: Second, what is the motivation for a decision-maker to aim for demographic parity (equal acceptance rates) in a medical treatment scenario, i.e., why is it the goal to provide the same proportion of service to blue and red patients?
>
> It is incorrect that the only goal of the decision-maker is to provide the same proportion of service. As clarified above, the decision-maker has its own reward to maximize (such as the percentage of healthy individuals over time), while trying to achieve demographic parity.
>
> > W1.3: The motivation of the proposed fairness notion lives from the observation that it would be “unfair” to provide nobody of the group of 100 red people with a treatment at time step t, just because we do not provide the 1 blue person at time step t with treatment.
>
> This is not our motivation. Our motivation is: in order to **equally** treat any red person at $t$ (with 100 red people) and the red person at $t+1$ (with 1 red people), we should not treat both timesteps equally.
>
> We refer the reviewer to our response to W3 for more detailed explanations.
>
> > W1.4: I am struggeling to think of a real-world scenario, where a decision-maker would take such decisions, thus I have difficulties to follow the motivation for the example and thus the proposed fairness notion. As indnicated in [S2], it would be easier for me to see an application of this in a hiring scenario.
>
> All the scenarios (lending, medical treatment allocation, attention allocation) that we considered in the paper, as well as the hiring example mentioned by the reviewer, are in fact the examples where our proposed notion are more preferable than prior notions. This is because in all these scenarios, individuals from the same group arriving at different timesteps should have the same impact on the bias computation (just like in static fairness notions every individual from the same group has the same impact on static fairnss metrics). **Our proposed metric satisfies this whereas prior notions do not**. We refer the reviewer to our response to W3 for more detailed explanations.

---

> ### Author Response · Authors · 2023-11-14
>
> ___
> > W2: If I understood the example and fairness notion correctly, the demographic parity notion from the introduction is only expected to yield a different result than static one-step fairness notions, when i) demographic group sizes vary across time steps and ii) if a) decisions are trivial (i.e., either “accept all” or “accept nobody”) or b) we have one group with a group size 1. Here is a scenario, where i) is fulfilled, but ii) not: When we have 20 non-binary individuals and 100 binary individuals at t, then 80 non-binary individuals and 10 binary individuals at t+1 and fulfill DP with an acceptance rate p \in (0, 1) at each time step (such that DP-unfairness is zero), then the proposed long term notion yields the same as the single time step notion, e.g., p_t = 0.5, p_{t+1} = 0.25, then |(10+40)/(20+80) – (50+5) / (100+10)| = |50/100 – 55/110| = |0|. To my understanding from the example, the notion is only useful in a very specific case, which limits its practicality. I am happy to be corrected by the authors, if I misunderstood.
>
> The computation provided by the reviewer is not correct. The provided example in fact shows that our proposed metric yields **different** results from prior notions most of the time, contrary to the reviewer's understanding.
>
> First, It seems that there is a mistake in your example: $p_{t+1} = 0.5$, not 0.25. This is because $\frac{40}{80}$ = $\frac{5}{10}$ = 0.5.
>
> Second, this is actually a good example so let's do some calculation using $p_t$ and $p_{t+1}$.
> 1. In the provided example, the step-wise biases are zero for both time steps. So the question is whether our proposed notion also yields zero bias.
> 2. In our proposed notion: $|(20p_t+80p_{t+1})/(20+80) – (100p_t+10p_{t+1}) / (100+10)| = \frac{78}{110}|p_t - p_{t+1}|$. Therefore, our notion computes zero bias if and only if $p_t = p_{t+1}$ in your example. For example, it will compute zero bias when $p_t = p_{t+1} = 0.5$. However, in general $p_t$ and $p_{t+1}$ are different and thus our notion will yield different results than prior notions.
>
> Therefore, the correct conclusion is: the proposed notion is expected to yield a different result from static one-step notions in general. They yield the same results only in very specific cases (e.g. $p_t = p_{t+1}$ in the provided example).
>
>
> ---
> > W4: The authors claim on page 2: “Without considering the variation in temporal importance within a group, these prior metrics lead to a false sense of fairness.” I strongly object the attempt to claim that prior fairness notions are “false”, which the authors do several times on page 2. Different fairness notions capture different understandings of fairness and which one to pick is very dependent on the context and scenario, and importantly should be at the heart of different research fields, such as philosophy, ethics, sociology etc. I advocate to remove these strong claims....
>
> First we encourage the reviewer to carefully read our response to W3.
>
> It is a misunderstanding that our paper `"attempts to claim that prior fairness notions are “false”"`. We agree that different fairness notions capture different understandings of fairness. However we did identify how prior notions can lead to a false sense of fairness: they can cause discrimination since individuals arriving at different timesteps have distinct impact on the bias metrics. This is concerning in many real-world applications (e.g. lending, medical treatment allocation, attention allocation and hiring example provided by the reviewer).
>
> ---
> > W5: In the introduction, the authors keep writing that they measure “bias”, but to my understanding, what they are measuring is not a (statistical, sampling, cofounding etc.) bias, but their moral/ethical understanding of “unfairness”.
>
> This is a misunderstanding of our terminology. The "bias" in our introduction is in fact a statistical concept (difference in the long-term benefit rate).
>
> Also, it is not true that our definition just come from our `moral/ethical understanding of “unfairness”`. In fact, long-term fairness metrics of all environments used in our experimental section were proposed and have been used in prior works, e.g. [1,2,3]. These metrics actually fall under ELBERT's general framework instead of prior notions (e.g. naively summing of step-wise biases). This should serve as a very strong motivation for using our proposed ELBERT.
>
>
>
> Ref:
> [1] D’Amour, Alexander, et al. “Fairness is not static: deeper understanding of long term fairness via simulation studies.” Proceedings of the 2020 Conference on Fairness, Accountability, and Transparency. 2020.
> [2] Yu, Eric Yang, et al. “Policy Optimization with Advantage Regularization for Long-Term Fairness in Decision Systems.” Neurips 2022.
> [3] Atwood, James, et al. "Fair treatment allocations in social networks." arXiv preprint arXiv:1911.05489 (2019).

---

> ### Author Response · Authors · 2023-11-14
>
> ---
> > W6: In the conclusion the authors write “...extra constraints on demand might be needed. For example, the demand should not be too large (e.g. when demand is the number of infected individuals) or too small (e.g. when demand is the number of qualified applicants).” ...“Demand” in how the users understand it, is the number of individuals a decision-maker is deciding upon at each time step (e.g., loan applicants). I am failing to see, how and why these extra constraints might be needed.
>
> In the conclusion we said that extra constraints on **group demand** might be needed. Here is an example:
>
> In medical treatment allocation, where the demand is defined as the number of individuals arriving at hospital. If the demand is very low, it could mean people are healthy so they do not go to hospital. Or it could mean people are not willing to go to hospital, which is potentially a bad thing. Depending on the definition of demand and the real world situation, extra constraints on group demand might be needed.
>
> ---
> > W7: There are missing citations for the background section, e.g., for Standard MDP. In the experimental section, there are missing citations for each of the three environments considered.
>
> Thanks for pointing out!
> 1. Standard MDP:  We have added citations for this.
> 2. Environments: We have edited the first paragraph in section 5.1 for more proper citation.
> ---
> > (Minor) In times of increasing research on large language models, I would suggest the authors to pick a different name for their fairness notion than “ELBERT”, given the similarity to all the variations of BERT models.
>
> Thanks for pointing out the similarity to BERT models! We will consider changing a different name for the notion.
>
> ---
> > Question: The authors write on page 3 “...the formulation of the group well-being can be unified as the ratio between supply and demand.“ I have difficulties to understand, where this understanding of demand and supply in fairness notions comes from. There is no citation given. To my understanding, at a classical sequential decision making scenario... and the decision maker takes the features of the individuals as input and decides usually aiming to maximize their utility while potentially adhering to some fairness notions. The concept of supply and demand comes from economics, where the general standard assumption is that goods demanded/supplied are uniform, i.e., every individual item is the same. This is not the case in fair decision making; every individual is different. There is not a demand and supply of credits and as long as a decision maker can supply credits they will. Instead, they will aim to give the credit to the individuals that they expect to have a high probability of paying back. Could the authors please clarify this?
>
> 1. (Ratio, supply & demand) Since many popular static fairness notions (such as demographic parity, equal opportunity, etc) involve comparing ratios between groups (such as acceptance rate), we adopt the same mathemtical form of the ratio. We did cite these static notions in our paper. We name "group demand" and "group supply" ourselves for lack of better terms. In static/sequential fair decision making, although every individual is different, they are treated equally when computing the fairness metrics. Therefore, this terminology is still consistent with the ones in economics.
> 2. `Instead, they will aim to give the credit to the individuals that they expect to have a high probability of paying back.`
>
>     That is right! In the lending example, the bank aims to maximize profits so they will try to give the credit to the individuals that they expect to have a high probability of paying back. At the same time, the bank also needs to ensure fairness. Our proposed ELBERT-PO method helps maximize reward while reducing the bias.
>
> ---
> We appreciate the discussions with the reviewers on our proposed notions. We also encourage the reviewer to appreciate our methodology contribution on developing the policy optimization for bias mitigation in sequential decision making, and also our state-of-the-art bias mitigation performance.
>
> Note that all the environments and the specific bias criterion (fairness notions) in the experiments were also extensively used in prior works [1,2]. These fairness criterion are in fact consistent with our proposed ELBERT notion instead of prior notions.
>
> Ref:
> [1] D’Amour, Alexander, et al. “Fairness is not static: deeper understanding of long term fairness via simulation studies.” Proceedings of the 2020 Conference on Fairness, Accountability, and Transparency. 2020.
> [2] Yu, Eric Yang, et al. “Policy Optimization with Advantage Regularization for Long-Term Fairness in Decision Systems.” Neurips 2022.
>
> ---
>
> Thank you again for your time and effort in reviewing our paper! Please let us know if the above explanations do not address your concerns. We are happy to answer any further questions.

---

> ### Author Response · Authors · 2023-11-19
> **Does our response address your concerns?**
>
> Dear reviewer 1EzK,
>
> As the stage of the review discussion is ending soon, we would like to kindly ask you to review our revised paper as well as our response and consider making adjustments to the scores. Please let us know if there are any other questions. We would appreciate the opportunity to engage further if needed.
>
> Best regards,
>
> Paper1977 Authors

---

> > ### Author Response · Authors · 2023-11-21
> > **Does our response address your concerns?**
> >
> > Dear reviewer 1EzK,
> >
> > As the stage of the review discussion is ending soon, we would like to kindly ask you to review our revised paper as well as our response and consider making adjustments to the scores. Please let us know if there are any other questions. We would appreciate the opportunity to engage further if needed.
> >
> > Best regards,
> >
> > Paper1977 Authors

---

### Official Review · Reviewer_na9A · 2023-11-02

**Soundness:** 2 fair
**Presentation:** 3 good
**Contribution:** 2 fair
**Rating:** 5
**Confidence:** 4

**Summary:**

The paper considers long-term fairness through the ratio between aggregated group supply and demand. The paper proposes supply-demand Markov decision process (SD-MDP) to model group welfare in terms of the ratio between supply and demand, and proposes the ELBERT notion of long-term fairness based on static group-level fairness notions (EOdds and DP). Policy optimization approach is presented to achieve ELBERT.

**Strengths:**

Overall, the paper is not hard to follow. The authors present the setting, the modeling of the problem, and the proposed solution relatively clear. It is nice to see the attempt to draw connection and apply static fairness notions in dynamic scenarios.

**Weaknesses:**

The weakness of the paper comes from the not-clearly-motivated objective of interest in terms of the application of static fairness in long-term and dynamic settings. In particular, clarifications on points in Section __Questions__ would be helpful.

**Questions:**

__Question 1__: the applicability of the way long-term fairness is characterized

In the motivating example, the hospital dispatches 0 treatment (action) to both blue and red groups, and dispatches a total of 101 treatments (actions) to both groups. As noted by authors, the treatment rate is 0 at time t, and 1 at time t+1 for both groups. I am not sure this is a good motivating example to emphasize the relation between the resource distributed and the group-wise importance of the state. First of all, why the supplies are modeled in a group-specific way from the user (here, is red/blue group) perspective, instead of from the supplier (here, is the hospital) perspective? Second, while I understand authors' emphasis on the fact that ratio is calculated after aggregation, I do not think the proposed modeling is more general than aggregating the supply-demand ratios along the temporal axis. In fact, those two modeling choices are complimentary, each suitable for specific practical scenarios. Third, the proposed long-term fairness notion is not robust towards group size imbalance. Imagine that the group size of red is significantly larger than that of blue, to the extent that even for the most unimportant states of red group, the demand is much larger than the entire population of blue group (and therefore, larger than the demand of blue group even in its most important state). One can imagine that in the calculated ratio, the supply/demand of blue group does not alter the result simply because they are out-numbered to a certain extent. The proposed fairness evaluation, in turn, may not provide sensible distinction between numerical variation and suffering of minority group.

__Question 2__: the claimed benefit of the proposed notion, compared to previous notions

As a follow-up on previous question, the proposed notion is claimed to address potential issues of previous utilization of static fairness in dynamic settings. Specifically, authors claim that the sequence of applying aggregation and ratio calculation matters. It is true that states have relative importance for each group. The discussion should include more than one (somewhat exaggerating) example to establish the (potential) advantage of ratio-after-aggregation compared to ratio-before-aggregation. To me, these two ways are complimentary, and the proposed notion is not well-motivated enough.

__Question 3__: the "importance" of state vs. long-term fairness goal

If the importance corresponds closely to the absolute size of demand at a given time, it is does not seem natural to model group-specific importance of states for the purpose of achieving long-term fairness. After all, if the focus is on fairness of resource distribution, what is the implication of state importance on long-term fairness, when the importance is only an intrinsic property of a certain state (e.g., if the base population is large).

---

> ### Author Response · Authors · 2023-11-14
>
> We thank Reviewer na9A for the detailed and insightful feedback. We are encouraged that the reviewer finds our paper clear and appreciate our effort in drawing connection from static fairness notions to sequential settings. Below we address the reviewer's concerns in detail.
>
> ---
> >Q1.1: In the motivating example, ... First of all, why the supplies are modeled in a group-specific way from the user (here, is red/blue group) perspective, instead of from the supplier (here, is the hospital) perspective?
>
> The group supplies are modeled in a group-specific way from the red/blue groups' perspective, since "group supply" is for computing fairness metrics (e.g. allocation/acceptance rate). From the supplier's (hospital) perspective, supplies can be viewed as part of its action. Different choices of actions will result in different long-term reward and long-term bias.
>
> > Q1.2: Second, while I understand authors' emphasis on the fact that ratio is calculated after aggregation, I do not think the proposed modeling is more general than aggregating the supply-demand ratios along the temporal axis. In fact, those two modeling choices are complimentary, each suitable for specific practical scenarios.
>
> It is incorrect that prior notions are suitable for `"pratical scenarios"` considered in our paper (including lending, medical treatment allocation and attention allocation). In fact, prior notions have a concerning downside that: by first computing the ratio and ignoring temporal importance difference, they discriminate individuals from timesteps with higher group demand, which we clarify below.
>
> Under prior notions (first computing step-wise ratio, then aggregating), within a group, individuals from timesteps with larger group size have smaller impact on bias computation, causing discrimination. To see this, consider the red group in the motivating example again: at time $t$ the group size is 100, and at $t+1$ the group size is 1. For any individual from time $t$, the effect of the decision, changing from "rejection" to "acceptance" will change the step-wise ratio by only $\frac{1}{100}$. However, the effect of decision on step-wise ratio for any individual from $t+1$ is $\frac{1}{1}$, much larger than $\frac{1}{100}$. Therefore, when improving the red group's overall well-being under prior notions, the individual from smaller group size of 1 at $t+1$ is more favorable since the decision on this individual has larger impact. Individuals from $t$ are discriminated due to smaller impact when measuring fairness.
>
> On the other hand, under our notion (first aggregating over time, then computing the ratio), within a group, individuals from different timesteps have the **same** impact on bias computation. To see this, consider the red group in the motivating example: at time $t$ the group size is 100, and at $t+1$ the group size is 1. For any individual from either time $t$ or time $t+1$, the effect of the decision, changing from "rejection" to "acceptance" will change the Long-term Benefit Rate by $\frac{1}{101}$, which is the same for both $t$ and $t+1$.
>
> Therefore, our proposed notion is more suitable than prior notions in practical scenarios.
>
> > Q1.3: Third, the proposed long-term fairness notion is not robust towards group size imbalance. Imagine that the group size of red is significantly larger than that of blue... One can imagine that in the calculated ratio, the supply/demand of blue group does not alter the result simply because they are out-numbered to a certain extent. The proposed fairness evaluation, in turn, may not provide sensible distinction between numerical variation and suffering of minority group.
>
> It is incorrect that `"the proposed long-term fairness notion is not robust towards group size imbalance"`and `"the supply/demand of blue group does not alter the result simply because they are out-numbered"`. This is because computing the long-term benefit rate for a group, only involves taking the ratio between the cumulative group supply and demand **for this group**. The size of another group do not affect the computation of long-term benefit rate for this group. The bias is then computed as the differences of long-term benefit rate between different groups.
>
> As an example, say the demand of red group is 100 at $t$ and 200 at $t+1$, where as the demand of blue group is 2 at $t$ and 1 at $t+1$. The supply of red group is 10 at $t$ and 20 at $t+1$, where as the supply of blue group is 1 at $t$ and 1 at $t+1$. Overall the blue group is much smaller.
>
> The group benefit rate of blue group is $\frac{1+1}{2+1}=\frac{2}{3}$, which is not affected by the fact that the group size (demand) of the red group is much larger than blue group.
>
> In this example, the long-term benefit rate of red group is $\frac{10+20}{100+200}=\frac{1}{10}$. The bias is computed as the difference in benefit rates, which is $\frac{2}{3}-\frac{1}{10}$. In this case, the blue group with smaller population size is actually the advantaged group.

---

> ### Author Response · Authors · 2023-11-14
>
> ---
> > Q2: The claimed benefit of the proposed notion, compared to previous notions; As a follow-up on previous question, the proposed notion is claimed to address potential issues of previous utilization of static fairness in dynamic settings. Specifically, authors claim that the sequence of applying aggregation and ratio calculation matters. It is true that states have relative importance for each group. The discussion should include more than one (somewhat exaggerating) example to establish the (potential) advantage of ratio-after-aggregation compared to ratio-before-aggregation. To me, these two ways are complimentary, and the proposed notion is not well-motivated enough.
>
> It is incorrect that `"these two ways are complimentary"` since prior notions have some serious downsides. We encourage the reviewer to carefully read our previous response to Q1.2.
>
> Although we use a `"somewhat exaggerating example"` in our paper, the following fact is general: prior notions can lead to a false sense of fairness/unfairness when different timesteps have different group demand, which is true in sequential decision making.
>
> Also, we would like to mention that the long-term fairness metrics of all environments used in our experimental section were proposed and have been used in prior works, e.g. [1,2,3]. These metrics actually fall under ELBERT's general framework instead of prior notions (e.g. naively summing of step-wise biases). This should serve as a very strong motivation for using our proposed ELBERT.
>
>
>
> Ref:
> [1] D’Amour, Alexander, et al. “Fairness is not static: deeper understanding of long term fairness via simulation studies.” Proceedings of the 2020 Conference on Fairness, Accountability, and Transparency. 2020.
> [2] Yu, Eric Yang, et al. “Policy Optimization with Advantage Regularization for Long-Term Fairness in Decision Systems.” Neurips 2022.
> [3] Atwood, James, et al. "Fair treatment allocations in social networks." arXiv preprint arXiv:1911.05489 (2019).
>
> ---
> > Q3: the "importance" of state vs. long-term fairness goal; If the importance corresponds closely to the absolute size of demand at a given time, it is does not seem natural to model group-specific importance of states for the purpose of achieving long-term fairness. After all, if the focus is on fairness of resource distribution, what is the implication of state importance on long-term fairness, when the importance is only an intrinsic property of a certain state (e.g., if the base population is large).
>
> 1. In the paper we demonstrate that, for a certain group, it is crucial to  consider the **relative** importance (demand) among timesteps, not the absolute size of demand. For example, even if `"the base population is large"`, the group demand still varies over time.
> 2. In sequential decision making, the current action has future effects. In particular, different actions will lead to different change in demand across timesteps. For example, some actions might result in increase in future demands.
> 3. While it is true that ```importance is a property of a certain state```, it is also important to note that in sequential settings, different actions will lead to different state transition in the future.
> 4. The proposed fairness metrics depend on supply and demand at all timesteps. Therefore, in order to achieve long-term fairness, one needs to deal with time-changing supply and demand when choosing the actions.
>
> ---
>
> Thank you again for your time and effort in reviewing our paper! Please let us know if the above explanations do not address your concerns. We are happy to answer any further questions.

---

> ### Author Response · Authors · 2023-11-19
> **Does our response address your concerns?**
>
> Dear reviewer na9A,
>
> As the stage of the review discussion is ending soon, we would like to kindly ask you to review our revised paper as well as our response and consider making adjustments to the scores. Please let us know if there are any other questions. We would appreciate the opportunity to engage further if needed.
>
> Best regards,
>
> Paper1977 Authors

---

> > ### Author Response · Authors · 2023-11-21
> > **Does our response address your concerns?**
> >
> > Dear reviewer na9A,
> >
> > As the stage of the review discussion is ending soon, we would like to kindly ask you to review our revised paper as well as our response and consider making adjustments to the scores. Please let us know if there are any other questions. We would appreciate the opportunity to engage further if needed.
> >
> > Best regards,
> >
> > Paper1977 Authors

---

> > > ### Comment · Reviewer_na9A · 2023-11-21
> > > **Follow-up Questions**
> > >
> > > Thank authors for the detailed responses. Below please see follow-up questions:
> > >
> > > **Regarding Response to Q1.1**
> > >
> > > I understand the fact that "'group supply' is for computing fairness metrics". But how one would like to evaluate fairness violation should not alter what counts as reasonable when modeling the dynamic process (modeling is the first step, after which one evaluates). If "supplies [from the hospital] are viewed as part of its action", then why the demand and supply (from red/blue group) are within the same state of the current step? Shouldn't it be demand (but not supply) as state and supply as action? Also, if there is no explicit assumption on the amount of supply from the hospital perspective, why there is insufficient supply for each group?
> > >
> > > **Regarding Responses to Q1.2 and Q1.3, and Q2**
> > >
> > > Thanks for providing an additional example. Let us use this example to evaluate two kinds of strategies. If we consider `first aggregation then ratio`, then the blue group is advantaged (since 2/3 > 1/10), as discussed by authors. If we consider `first ratio then aggregation`, then **still** the blue group is advantaged (since 75% > 10%). The two approaches come to the same conclusion. This is aligned with my comment that "these two ways are complimentary", and that each may be suitable for specific practical scenarios.
> > >
> > > Since the claimed benefit of the propose notions only exist in a relatively exaggerated example (acknowledged by authors), and that the newly provided example does not seem to support the claimed benefit, I am not convinced that such advantage is general.
> > >
> > > Furthermore, the authors claim that "the following fact is general: prior notions can lead to a false sense of fairness/unfairness when different timesteps have different group demand." If the difference in group demand (but not supply) is the key, then why not just model supply as an action (this point is connected to `Regarding Response to Q1.1`)?

---

> ### Author Response · Authors · 2023-11-22
>
> Thank you for your follow-up questions!
>
> > A.1: Regarding Response to Q1.1. I understand the fact that "'group supply' is for computing fairness metrics". But how one would like to evaluate fairness violation should not alter what counts as reasonable when modeling the dynamic process (modeling is the first step, after which one evaluates). If "supplies [from the hospital] are viewed as part of its action", then why the demand and supply (from red/blue group) are within the same state of the current step? Shouldn't it be demand (but not supply) as state and supply as action?
>
> Your understanding is correct and is consistent with our full descriptions of this medical treatment allocation problem in Section 5.1, Case 2. Specifically,
>
> * Decision-maker: the hospital.
> * State space of the MDP includes (1) the health condition of the whole population (which determines the group **demand**); (2) The condition of the hospital (for example, the number of medicine available at that time step).
> * Action space of the MDP includes whether to allocate medicine to each individuals, which determines group **supply**.
>
> The supply is not included in the state space and is determined by actions. Note that technically supply is not the same as action, but is determined by action: action is whether the hospital allocates treatment to each individual, while supply is the sum of all allocated treatments.
>
> Now that we have defined the MDP (we are done with modeling), the long-term benefit rate (for fairness evaluation) is defined to be the ratio between cumulative supply and demand, which is related to both state and actions.
>
> > A.2: Also, if there is no explicit assumption on the amount of supply from the hospital perspective, why there is insufficient supply for each group?
>
> Since this is a resource allocation problem, there is an implicit assumption that the supply is not unlimited. In Section 5.1 and Appendix D.1, we fully describe the details of this environment. In this particular environment, the supply is actually 1 at any given step. We encourage you to read our description in Appendix D.1 for more details.
>
>
> > B.1 .. additional example. If we consider first aggregation then ratio, then the blue group is advantaged (since 2/3 > 1/10), as discussed by authors. If we consider first ratio then aggregation, then still the blue group is advantaged (since 75% > 10%). The two approaches come to the same conclusion. This is aligned with my comment that "these two ways are complimentary", and that each may be suitable for specific practical scenarios. Since the claimed benefit of the propose notions only exist in a relatively exaggerated example (acknowledged by authors), and that the newly provided example does not seem to support the claimed benefit, I am not convinced that such advantage is general.
>
> We would like to point out that the "first ratio then aggregation" approach (prior approach) has a concerning downside, and thus should not be used in all application cases considered in our paper. The downside is: prior approach **discriminates** against individual coming from a timestep with higher demand. This is explained in our global response and we explain below using the example you refer to.
>
> Consider the red group. If the supply of an individual at time $t$ change from 0 to 1, then the group benefit computes as $(\frac{10+1}{100} + \frac{20}{200})/2 = 0.105$. If the supply of an individual at time $t+1$ change from 0 to 1, then the group benefit computes as $(\frac{10}{100} + \frac{20+1}{200})/2 = 0.1025 < 0.105$. Therefore, when using this approach, the individual at time $t$ is prefered because the decision on this person leads to larger increase in group benefit. However, in real world application individuals arriving at different timesteps should be treated equally.
>
> Therefore, these two fairness notions are not complimentary. They might give the same result on which group is more advantaged, but the "first ratio then aggregation" notion causes discrimination when one wants to improve fairness according to it. Our newly provided example does support our claim.
>
>
> > B.2 Furthermore, the authors claim that "the following fact is general: prior notions can lead to a false sense of fairness/unfairness when different timesteps have different group demand." If the difference in group demand (but not supply) is the key, then why not just model supply as an action (this point is connected to Regarding Response to Q1.1)?
>
> See our response to A.1. The supply is indeed determined by action but technically is not the same as action.

---

### Author Response · Authors · 2023-11-14
**ELBERT treats individuals from different timesteps equally, whereas prior notions do not**

We thank all reviewers for their valuable feedback. We are encouraged that the reviewers find fairness in sequential settings interesting and important (1EzK,na9A) and our paper well-written (na9A, aJ3j). Below we address some common confusion.

---
## ELBERT treats individuals from different timesteps equally

In our paper, we underscore the importance of recognizing that certain timesteps hold greater significance than others. Nevertheless, there have been misconceptions regarding our proposed ELBERT framework. Contrary to interpretations suggested by Reviewer 1EzK, the framework does not disproportionately prioritize individuals from larger groups. Similarly, the assertion by Reviewer na9A, that the importance of the timestep is solely tied to the base population, is a misunderstanding of our methodology.

We clarify that our ELBERT framework equally considers individuals across varying timesteps, irrespective of the differing group sizes at each step. Contrary to previous approaches that tend to de-emphasize the significance of individuals in timesteps with larger groups, leading to discriminatory outcomes, our method maintains equitable treatment. This is further analyzed below.

- **(a) Prior notions do not treat individuals equally from different timesteps**.
    Prior methodologies exhibit an unequal treatment of individuals across different timesteps. Specifically, these approaches calculate the step-wise ratio first and then aggregate it. This calculation method results in a skewed impact of decisions on individuals from larger group sizes in terms of bias computation.

    For instance, consider the red group in our motivating example. At time $t$, this group has 100 members, and at $t+1$, it reduces to just one member. Under these conditions, a decision shift from "rejection" to "acceptance" for an individual at time $t$ alters the step-wise ratio marginally, by $\frac{1}{100}$. In stark contrast, the same decision change for an individual at $t+1$ significantly affects the step-wise ratio, by $\frac{1}{1}$, which is considerably more impactful than the change at $t$.

    Consequently, when aiming to enhance the overall well-being of the red group, the individual at the smaller group size of 1 at $t+1$ becomes disproportionately influential, as decisions regarding this individual have a greater effect. This leads to a discriminatory outcome where individuals at time $t$, due to their minimal impact on fairness measures, are effectively marginalized.

     Moreover, in the red group example, all 100 individuals at time $t$ face rejection, with each of these decisions having a reduced weight compared to the acceptance decision at time $t+1$. Therefore, this approach of computing step-wise ratios before aggregation inherently creates biases against individuals from larger group sizes at certain timesteps. It presents a fundamental issue in the real-world scenarios we examine in our paper, such as medical treatment allocation, bank lending, and attention allocation. In these contexts, such a disproportionate impact of decisions on smaller groups at individual timesteps can lead to skewed and potentially unfair outcomes, undermining the integrity of the decision-making process in these critical applications.

- **(b) Our proposed ELBERT treats every individual equally**.
    In our ELBERT framework, we address the issue of unequal treatment across timesteps by adopting a novel approach for each group. ELBERT first aggregates the supply and demand over all timesteps and then calculates the ratio. This method ensures equal treatment of every individual, regardless of the timestep they belong to.


    To illustrate this, let's revisit the red group in our motivating example: At time $t$, this group has 100 members, and at $t+1$, the number dwindles to just 1. In the ELBERT framework, the impact of a decision change for any individual—whether from time $t$ or $t+1$  ---  from "rejection" to "acceptance" uniformly alters the Long-term Benefit Rate by $\frac{1}{101}$. This means that the decision's impact on the Long-term Benefit Rate is consistent and equal for individuals at both $t$ and $t+1$, demonstrating ELBERT's capacity to treat individuals from different timesteps with the same level of importance.

- **($\textbf{c}$) Treament of temporal difference in sequential decision-making**.
    In prior notions, there's a disproportionate emphasis on the impact of decisions based on group size across different timesteps when computing bias. Prior approach treats a timestep $t$ with a large group size (e.g., 100) equivalently to a timestep $t+1$ with a significantly smaller group size (e.g., 1). Consequently, the decision affecting a single individual at $t+1$ exerts a substantially larger influence on the fairness metric, which can be flawed in real-world applications.

---

> ### Author Response · Authors · 2023-11-14
> **The motivating example involves both reward and fairness**
>
> ---
> ## The motivating example involves both reward and fairness
>
> We wish to provide clarifications regarding the medical treatment allocation problem discussed in our introduction, where there is misunderstanding among reviewers (1EzK and na9A).
>
> 1. This example features as one of the experimental environments, detailed in Case 2, Section 5.1.
> 2. Excluding fairness considerations, this problem represents a typical sequential decision-making scenario where the objective is to maximize the reward, defined as the overall percentage of a healthy population over time.
> 3. With fairness in mind, the decision-maker aims to minimize bias, adhering to the principle of demographic parity. The notions of "group demand" and "group supply" are instrumental in bias measurement, viewed from individual perspectives, as fairness is evaluated at this level.
> 4. In our initial writing, we prioritized brevity and focused solely on fairness in the introduction, inadvertently omitting the reward aspect. This omission led to some confusion.
>
> To eliminate this ambiguity and reaffirm that our sequential decision-making framework encompasses both reward maximization and bias minimization, we have revised the introduction. It now features a Bank Lending example, which may be more relatable. The revisions include: (1) Defining the problem as a sequential decision-making process where the bank seeks to maximize long-term profits (the reward), and (2) acknowledging the bank's consideration of bias. (3) Additionally, the main figure in the introduction now explicitly represents both the reward and fairness-related elements.
>
> ## Our Methodological Contributions
>
> We are grateful for the insightful discussions on our novel approach in extending existing static fairness notions to their dynamic, long-term counterparts. We encourage the reviewers to recognize the significant advancements our research offers in the realm of policy optimization for bias mitigation within sequential decision-making contexts. Additionally, our methodology demonstrates state-of-the-art performance in mitigating bias.
>
> ---
>
> Thank you again for your effort in reviewing our paper! We are happy to answer any further questions.

---

### Meta-Review · Area_Chair_JKju · 2023-12-09

**Metareview:**

The paper introduces the concept of long-term fairness by examining the ratio between aggregated group supply and demand. It proposes a supply-demand Markov decision process (SD-MDP) to model group welfare based on this ratio and introduces the ELBERT notion of long-term fairness, which relies on static group-level fairness notions (EOdds and DP). The paper also presents a policy optimization approach to achieve ELBERT, emphasizing a novel perspective on group size that considers all individuals from a sensitive/demographic group across time steps rather than determining it at each time step, and outlines how to learn a policy adhering to this proposed fairness notion in a Markov Decision Making process setting.

While all reviewers applaud the clarity of the paper, the paper suffers from unclear motivation of the proposed notion of study - a couple of reviewers suggested more clear motivating examples and efforts to explain them. In addition, the technical contribution of the paper is relatively marginal, other than the newly defined fairness notation, which is okay but then the requirement for justifying the relevance of the fairness is higher. The paper can also benefit from a more careful revision of several definitions, for example, “bias” is a relatively overloaded term and the authors might consider better define it;  the authors are also encouraged to better position the role of their fairness measure with the existing static and long-term ones.

**Justification For Why Not Higher Score:**

As detailed in the meta review, the paper can benefit from another round of revision to better motivate their proposed fairness notation and justify its relevance. The authors are also encouraged to revise some definitions and wordings.

**Justification For Why Not Lower Score:**

N/A

---

### Decision · Program_Chairs · 2024-01-16

Reject